# BasisFormer: Attention-based Time Series Forecasting with Learnable and Interpretable Basis

**Zelin Ni**[*]
Shanghai Jiao Tong University
Shanghai, China
nzl5116190@sjtu.edu.cn

**Hang Yu**[*]
Ant Group
Hangzhou, China
hyu.hugo@antgroup.com

**Shizhan Liu**
Shanghai Jiao Tong University
Shanghai, China
shanluzuode@sjtu.edu.cn

**Jianguo Li**[†]
Ant Group
Hangzhou, China
lijg.zero@antgroup.com

**Weiyao Lin**[†]
Shanghai Jiao Tong University
Shanghai, China
wylin@sjtu.edu.cn

## Abstract

Bases have become an integral part of modern deep learning-based models for time series forecasting due to their ability to act as feature extractors or future references. To be effective, a basis must be tailored to the specific set of time series data and exhibit distinct correlation with each time series within the set. However, current state-of-the-art methods are limited in their ability to satisfy both of these requirements simultaneously. To address this challenge, we propose BasisFormer, an end-to-end time series forecasting architecture that leverages learnable and interpretable bases. This architecture comprises three components: First, we acquire bases through adaptive self-supervised learning, which treats the historical and future sections of the time series as two distinct views and employs contrastive learning. Next, we design a Coef module that calculates the similarity coefficients between the time series and bases in the historical view via bidirectional cross-attention. Finally, we present a Forecast module that selects and consolidates the bases in the future view based on the similarity coefficients, resulting in accurate future predictions. Through extensive experiments on six datasets, we demonstrate that BasisFormer outperforms previous state-of-the-art methods by 11.04% and 15.78% respectively for univariate and multivariate forecasting tasks. Code is available at: https://github.com/nzl5116190/Basisformer.

## 1 Introduction

Bases, such as trends and seasonalities, are indispensable for time series modeling and forecasting, since they capture the underlying temporal patterns and serve as the key factors driving changes in the data over time. For example, seasonalities can capture the regular fluctuations in demand for a product or a service, while trends can reflect the long-term growth or decline of a market or industry. Incorporating these bases into time series models can improve the understanding and prediction of future behaviors. Indeed, all commonly used deep learning models for time series forecasting can be reimagined as basis-driving models. N-BEATS [1], N-HiTS [2], and FiLM [3] all resort to explicit bases, such as Fourier and Legendre basis. More generally, the linear (or convolution) layers in MLPs [4] (or CNNs [5]) can be regarded as implicit bases, as they act as filter banks to decompose the time series. In addition, the covariate embedding (a.k.a. global timestamp embedding) in RNNs [6]

---

[*]Equal contribution. This work was done when Zelin Ni was a research intern at Ant Group.
[†]Corresponding authors.

37th Conference on Neural Information Processing Systems (NeurIPS 2023).

and Transformers [7–13] is another form of basis, since they provide reference points for predicting future sequences.

To apply bases for time series forecasting, three steps are required. First and foremost, an appropriate basis should be chosen or learned for the set of time series at hand. In practice, the full space of the basis can often be very large, whereas the patterns of the time series in a set are often similar. It is therefore desirable to learn a basis that is tailored to the specific characteristics (e.g., periods or frequencies) of the time series data. This can help reduce the complexity of the forecasting model, as well as make it more accurate and interpretable. Second, each time series in the set is decomposed according to the basis. This involves computing coefficients or weights that determine the similarity or projection energy of the time series w.r.t. (with respect to) each vector (i.e., filter) in the basis. Note that these coefficients are supposed to vary across different time series, since each time series also exhibits unique patterns and characteristics. For instance, the Fourier coefficients corresponding to different time series will be dissimilar. Finally, the prediction is determined by the weighted aggregation of the future part of the basis.

Unfortunately, the aforementioned state-of-the-art methods fall short in satisfying the requirements in the first two steps simultaneously. On one hand, methods relying on classical bases, such as N-BEATS [1], N-HiTS [2], and FiLM [3], often assume that the basis is not learnable and instead learn the coefficients of each time series in a flexible manner when projecting to the basis. However, such a basis may not effectively account for temporal patterns since there is no guarantee that the given basis includes all periods or frequencies corresponding to the set of time series. On the other hand, approaches that aim to adaptively learn the basis from data, such as MLP [4], CNN [5], RNN [6], Transformer and their variants [7–13], often overlook the need for flexible associations between the basis and each individual time series. Specifically, although Transformer and its variants learn the covariate embeddings, they add or concatenate the same embeddings to different time series embedding in a restricted manner. For MLP and CNN, they adopt the same linear and convolution layers for all time series.

To effectively tackle the aforementioned quandary, it is imperative to obtain a basis that can accurately reflect the distinct traits of the dataset, and to devise a predictive network that can selectively utilize relevant vectors in the basis for forecasting purposes. To move forward to this goal, we propose BasisFormer, a time series forecasting architecture with learnable and interpretable basis. As a first step, we acquire the basis through adaptive self-supervised learning from the data. This involves treating the historical and future sections of a time series as two distinct views and employing contrastive learning to learn the basis, assuming that the selection of basis for a time series should be consistent across both views. Subsequently, we design the Coef module that measures the similarity between the time series and the basis in the historical view via bidirectional cross-attention, facilitating the flexible association between individual time series and the basis. Finally, we develop a Forecast module that consolidates vectors from the basis in the future view according to the similarity yielded by the Coef module, leading to accurate future predictions. We emphasize that the above three parts are trained in an end-to-end fashion. In summary, the key contributions of our work comprise:

- We propose a self-supervised method for basis learning by treating the historical and future sections of the time series as two distinct views and employing contrastive learning, which ensures that the selection of basis for a time series is consistent across both views.

- We design the Coef and Forecast module that chooses and merges relevant basis in the future view based on the coefficients that measure the similarity between the time series and the basis in the historical view.

- We conduct extensive experiments on six datasets, and find that our model outperforms previous SOTA methods by 11.04% for univariate forecasting tasks and 15.78% for multivariate forecasting tasks.

## 2 Related works

**Time series forecasting models** In recent years, deep learning methods have emerged as the predominant technique for time series forecasting. As illustrated in the introduction, these deep learning methods typically resort to bases to facilitate the prediction of the future. Depending on the types of bases used in the network, forecasting models fall into two categories: those using classical orthogonal bases and those using learnable bases. The first group involves N-BEATS [1], N-HiTS [2], and FiLM [3]. N-BEATS and N-HiTs typically utilize the Fourier basis and then learn the coefficients

for this basis in a recursive network such that the basis helps decompose the historical part of the time series into different components and these components can be further aggregated to predict the future. FiLM approximates the historical part via the Legendre polynomial basis and further removes the noise with the help of the Fourier basis. The major drawback with the group of methods is that the basis is predefined, thus giving rise to the problem of which type of basis to choose (e.g., Fourier or Legendre polynomial) and further which vectors in the basis to choose (e.g., which frequency components we choose from the Fourier basis). On the other hand, the models based on learnable bases, such as Dlinear [4], TCN [5], Deepar [6], LogTrans [11], Informer [7], AutoFormer [8], FedFormer [9], etc, use a learnable linear or convolution layer, or covariate embeddings as the basis. Although these bases are adaptable to the time series, the relationship between the basis and the time series is fixed for all time series. For example, the covariate embedding is added or concatenated to the embedding of different time series in the same way, without considering the unique frequencies and periodic patterns of each series. In our paper, we propose a method that allows for both a learnable basis and a flexible correlation between the basis and each time series for more accurate predictions.

**Basis learning for time series analysis** Apart from time series forecasting, learnable bases have also been explored for other time series-related tasks, such as time series classification. Note that basis learning is distinct from the representation learning of time series, as the former aims to capture the pattern of a set of time series using a common basis, while the latter aims to extract features from individual time series. Moreover, the basis can help extract features from time series as in Dlinear [4] and TCN [5]. Traditionally, a non-learnable basis is exploited, such as Fourier and wavelet basis. However, there exist a handful of works that overcome this limitation and enable the use learnable basis. The learnable group transform [14] generalizes the filter banks in the wavelet transformer and allows nonlinear transformation from the mother wavelet to obtain flexible filter banks that can better extract features from time series. Along this direction, Balestriero *et al.* [15] propose a learnable Gaussian filter over the Wigner-Ville transform with a few interpretable parameters, and prove that the resulting filter banks can interpolate between classical ones, including Fourier, wavelet, and chirplet basis. Similar works have also been proposed for audio signal processing [16, 17], suggesting that a learnable basis is a more effective feature extractor than a fixed basis. Thus, we utilize a learnable basis in our work and demonstrate its usefulness for time series forecasting. It should be noted that DEPTS [18] tackles the challenges posed by intricate dependencies and multiple periodicities in periodic time series through the implementation of a deep expansion learning framework. However, the complex initialization and optimization strategies employed by DEPTS, as well as its limitation of only being applicable to periodic sequences, have motivated us to develop a simpler and more universally applicable basis learning framework. Concretely, we propose a novel method for basis learning based on self-supervised contrastive learning.

**Self-supervised time series representation learning** Since we employ self-supervised representation learning techniques to learn the basis, it is necessary to examine related works in this domain. One prevalent method in this field is the contrastive predictive coding (CPC) [19] which implements contrastive learning to obtain time series representations by treating subsequent future time series as positive samples and random non-subsequent future time series as negative samples. Another method, TS-TCC [20], augments the data with two types of perturbations to obtain two views and performs a cross-view prediction task contrastively, similar to CPC. As an alternative, TS2VEC [21] generates positive samples via timestamp masking or random cropping without leveraging the future information. Note that all these methods seek to establish a common representation for a time series from various views. Unlike these approaches, our goal is to preserve the consistency of the relationship between the time series and the basis. In other words, while the representation of the time series in the historical and future view may differ, their relationships with the corresponding basis should remain consistent.

## 3   BasisFormer

Suppose we have a collection of time series with a dimension of $C$, implying that $C$ correlated time series require simultaneous prediction. Each time series in this set is characterized by its history $\boldsymbol{x} = (x_1, \cdots, x_I)$ and future $\boldsymbol{y} = (y_1, \cdots, y_O)$, where $I$ and $O$ correspond to the input and output sequence lengths, respectively. Our primary objective is to learn a basis $\boldsymbol{z}$ that can account for the behavior of all time series in the group, and further exploit it for predicting $\boldsymbol{y}$ given $\boldsymbol{x}$. Correspondingly, $\boldsymbol{z}$ can also be split into the historical component $\boldsymbol{z}_x$ and the future component $\boldsymbol{z}_y$.

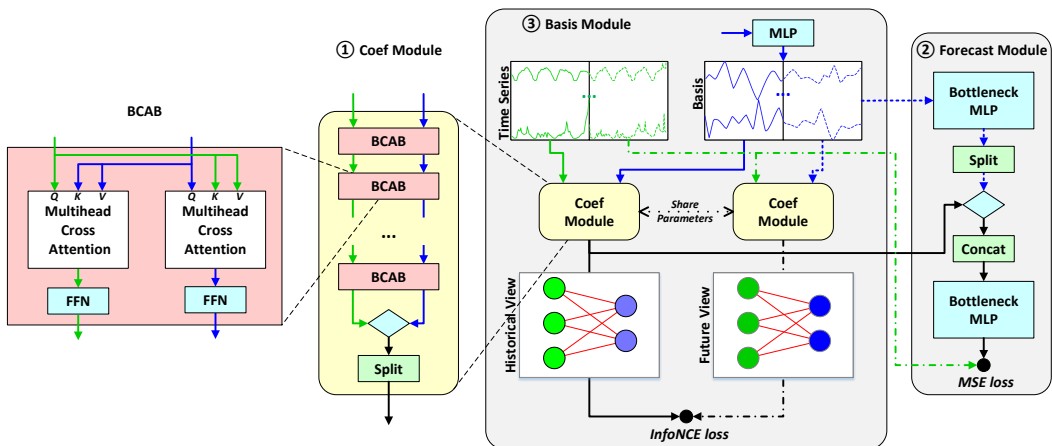

Figure 1: The architecture of BasisFormer, consisting of ① the Coef module, ② the Forcast module, and ③ the Basis module. The green and blue lines denote the data flow of the set of time series and basis vector repespetively. The cyan diamond denotes tensor dot product. Note that the dot-dash line, which denotes the data flow of the future part of the time series, is only included during training but removed during inference.

As mentioned in the introduction, three steps are necessary to employ bases for time series forecasting: learning an appropriate basis, computing the coefficient of the time series with respect to each basis vector, and predicting based on a weighted aggregation of the future portion of the basis. The proposed BasisFormer facilitates these three steps in a highly versatile manner. As demonstrated in Figure 1, the overall architecture of BasisFormer also contains three parts: ① the Coef module, which compares a time series to each basis vector to determine the corresponding coefficient; ② the Forecast module, which predicts the future based on the coefficients and the future section of the basis; and ③ the Basis module, which learns the basis by aligning the relationships between the basis and the time series from the historical and future perspectives. We will now elaborate on each of the BasisFormer components.

### 3.1 Coef module for similarity comparison between time series and basis

The Coef Module is designed to measure the similarity between a set of time series and a set of basis vectors. Since our focus is on the relations between the two sets, rather than within each set, we exploit a bipartite graph to represent such relations, where the nodes in one set represent the time series (see green nodes in Figure 1) and the nodes in the other set represent the basis vectors (see blue nodes in Figure 1). As a result, the strength of the edge connecting two nodes in the graph is equivalent to the similarity coefficient between them (see red edges in Figure 1). To obtain the edge strength, we require representations of the nodes in the graph. We accomplish this by developing a bidirectional cross-attention block (BCAB) to learn the node representations via cross-attention, in a similar vein to the graph attention network [22].

As a first step, given two sets of inputs $a^{(i)}$ and $b^{(i)}$, let us define the cross attention block (CAB) as:

$$\hat{a} = \text{LayerNorm}(\text{MA}_H(a^{(i)}, b^{(i)}, b^{(i)}) + a^{(i)}), \tag{1}$$

$$a^{(i+1)} = \text{CAB}_H(a^{(i)}, b^{(i)}) = \text{LayerNorm}(\text{FFN}(\hat{a}) + \hat{a}), \tag{2}$$

where $\text{MA}_H$ denotes multihead attention with $H$ heads whose query is given by $Q = W_q a$, key by $K = W_k b$, and value by $V = W_v b$. By exchanging information mutually between $a^{(i)}$ and $b^{(i)}$, we can construct the BCAB as:

$$(a^{(i+1)}, b^{(i+1)}) = \text{BCAB}_H(a^{(i)}, b^{(i)}), \tag{3}$$

where

$$a^{(i+1)} = \text{CAB}_H(a^{(i)}, b^{(i)}), \quad b^{(i+1)} = \text{CAB}_H(b^{(i)}, a^{(i)}). \tag{4}$$

Note that the parameters in $\text{CAB}_H$ for computing $a^{(i+1)}$ and $b^{(i+1)}$ may differ, so as to capture the heterogeneity in the relations from $a^{(i)}$ to $b^{(i)}$ and from $b^{(i)}$ to $a^{(i)}$.

Correspondingly, given $C$ time series $\boldsymbol{x} \in \mathbb{R}^{C \times I}$ and the basis $\boldsymbol{z}_x \in \mathbb{R}^{N \times I}$ of size $N$, we can get their representations by stacking $M$ layers of $\mathrm{BCAB}_H$, namely, $\boldsymbol{x}^{(M)} \in \mathbb{R}^{C \times D_c \times H}$ and $\boldsymbol{z}_x^{(M)} \in \mathbb{R}^{N \times D_c \times H}$, where $D_c$ represents the hidden dimension for each head in $\mathrm{BCAB}_H$. Note that the cross attention is computed between the time series and the basis, instead of across time which is frequently found in Transformer based models [7, 13]. Additionally, the attention mechanism is used to allow for flexible associations between the time series and basis vectors. This approach ensures that each time series can selectively attend to the most relevant basis vectors, and likewise, each basis vector can selectively attend to the most relevant time series.

Finally, the Coef module calculates the "coefficient" of each time series w.r.t. each basis vector as the inner product of their representations $x^{(M)}$ and $z_x^{(M)}$ for each of the $H$ heads, resulting in the coefficient tensor $\boldsymbol{c} \in R^{C \times N \times H}$.

## 3.2 Forecast module for aggregation and future prediction

After obtaining the coefficients, we take advantage of them for the sake of forecasting. We begin by projecting the future part of the basis vectors $\boldsymbol{z}_y$ into a space where they can be linearly aggregated using the coefficients. As the Coef module computes the coefficients for $H$ heads, the projection of $\boldsymbol{z}_y$ should also have $H$ heads to maintain consistency. To this end, we employ a four-layer Multi-Layer Perceptron (MLP) with a bottleneck to map $\boldsymbol{z}_y \in \mathbb{R}^{N \times O}$ to $\hat{\boldsymbol{z}}_y \in \mathbb{R}^{N \times O}$, which is then split into $H$ heads, each with size $N \times (O/H)$, and is denoted as $\tilde{\boldsymbol{z}}_y \in \mathbb{R}^{N \times H \times (O/H)}$.

For each head, we aggregate the $N$ basis vectors by computing the coefficients weighted sum of $\tilde{\boldsymbol{z}}_y$ over the dimension of $N$, that is,

$$\tilde{\boldsymbol{y}}[i, h] = \sum_{j=1}^{N} \boldsymbol{c}[i, j, h] \tilde{\boldsymbol{z}}_y[j, h, :], \tag{5}$$

where $h \in \{1, \cdots, H\}$ denotes the head index, and the size of $\tilde{\boldsymbol{y}}$ is $C \times H \times (O/H)$.

Next, we concatenate the $H$ heads together and pass them through another four-layer MLP with a bottleneck, in order to exchange information between different heads. This is because different heads may have captured different aspects of the input sequence and the fusion MLP can help combine the information and improve the overall prediction performance.

It is noteworthy that the bottleneck layers in the above module are used to reduce the dimensionality of the input features before projecting them to a higher-dimensional space. This helps to reduce the computational complexity of the projection operation and prevent overfitting. Additionally, using a bottleneck layer can also assist in extracting more informative features by forcing the model to learn a compressed representation of the input, thus improving the prediction accuracy.

Finally, we compare the predicted values $\hat{\boldsymbol{y}}$ with the true values $\boldsymbol{y}$ via a Mean Squared Error (MSE) loss function, that is, $L_{\mathrm{pred}} = \mathrm{MSE}(\hat{\boldsymbol{y}}, \boldsymbol{y})$.

## 3.3 Basis module for basis learning

In this subsection, we present our approach for learning a data-driven basis in a self-supervised manner. The goal is to obtain a basis that satisfies three essential properties.

First, the relation between the basis vectors and the time series should be consistent across time, such that we can predict the future by combining the future part of the basis using the coefficients obtained from the historical part of the basis and the time series. Specifically, given a time series $(\boldsymbol{x}, \boldsymbol{y})$ and the corresponding basis $(\boldsymbol{z}_x, \boldsymbol{z}_y)$, the coefficients (i.e., edge strength) between the time series and the basis should be consistent across the historical view $(\boldsymbol{x}, \boldsymbol{z}_x)$ and the future view $(\boldsymbol{y}, \boldsymbol{z}_y)$. In other words, the relevance of a given time series to a particular basis vector in the historical view should be retained in the future view. To achieve this, we pass both $(\boldsymbol{x}, \boldsymbol{z}_x)$ and $(\boldsymbol{y}, \boldsymbol{z}_y)$ through the Coef module to get the coefficient tensor respectively for the two views, $\boldsymbol{c}_x$ and $\boldsymbol{c}_y$, both with size $C \times N \times H$. For each time series, we then perform contrastive learning by regarding the coefficient w.r.t. each basis vector in $\boldsymbol{c}_x$ as the anchor point, the coefficient w.r.t. the corresponding basis vector in $\boldsymbol{c}_y$ as the positive sample, and the coefficient w.r.t. the remaining basis vectors in $\boldsymbol{c}_y$ as the negative sample. We optimize the InfoNCE loss to maximize the mutual information between $\boldsymbol{c}_x$ and $\boldsymbol{c}_y$,

which is given by

$$L_{\text{align}} = -\frac{1}{CN} \sum_{i=1}^{C} \sum_{j=1}^{N} \log \frac{\exp(\boldsymbol{c}_x[i,j,:] \cdot \boldsymbol{c}_y[i,j,:]/\epsilon)}{\sum_{k=1}^{N} \exp(\boldsymbol{c}_x[i,j,:] \cdot \boldsymbol{c}_y[i,k,:]/\epsilon)}. \tag{6}$$

where $\epsilon$ represents the temperature used to adjust the smoothness of alignment distribution.

In addition, we require the basis to be interpretable, which means that we can gain insights into the underlying patterns captured by the basis. To achieve interpretability, we promote smoothness across time with a regularization term, that is,

$$L_{\text{smooth}} = \|\boldsymbol{z}\boldsymbol{S}\|_2^2, \tag{7}$$

where we concatenate the basis vectors for the historical and future views $(\boldsymbol{z}_x, \boldsymbol{z}_y)$ along the last dimension to form $\boldsymbol{z}$, and the smoothness matrix $\boldsymbol{S} \in \mathbb{R}^{(I+O) \times (I+O-2)}$ can be expressed as:

$$\boldsymbol{S} = \begin{bmatrix} 1 & -2 & 1 & & \\ & \ddots & \ddots & \ddots & \\ & & 1 & -2 & 1 \end{bmatrix}. \tag{8}$$

It is apparent by multiplying $\boldsymbol{z}$ with $\boldsymbol{S}$, we compute $\|\boldsymbol{z}[:,t-1] - 2\boldsymbol{z}[:,t] + \boldsymbol{z}[:,t+1]\|_2^2$, which is the curvature over time [23]. One advantage of using $\boldsymbol{S}$ is that the addition of a constant and a linear function of time makes the loss invariant. Therefore, the above smoothness loss can accommodate the change of the overall mean level as well as the linear trend.

Finally, the basis should be a function of the timestamp. As such, we develop a four-layer MLP with a skip connection between the input and the output of the second layer. The input to the network is the normalized timestamp associated with the first time point in the historical window. Suppose that the overall length of a time series in the dataset is $T$, then the normalized timestamp is defined as $\tau = t/T$, where $t \in \{0, \cdots, T-1\}$. The output of the networks is an $N \times (I+O)$ tensor, which is the basis for the current time window.

Overall, the loss we optimize can be expressed as:

$$L = L_{\text{pred}} + L_{\text{align}} + L_{\text{smooth}}. \tag{9}$$

We find that the performance of BasisFormer is robust to the weights in front of the terms in (9). Therefore, we set the weights to be one in all our experiments. Sensitivity analysis of the weights in the loss function can be found in the Appendix A.4.

## 4   Experiments

To assess the effectiveness of our model, we conducted comprehensive experiments on six datasets from real-world scenarios, using the experimental setup identical to that in [5, 7–9]. Below we summarize the experimental setup, datasets, models, and compared models.

**Experimental setup**: The length of the historical input sequence is maintained at 96 (or 36 for the illness dataset), whereas the length of the sequence to be predicted is selected from a range of values, i.e., $\{96, 192, 336, 720\}$ ($\{24, 36, 48, 60\}$ for the illness dataset). Note that the input length is fixed to be 96 for all methods for a fair comparison.

**Datasets**: The six datasets used in this study comprise the following: 1) ETT [7], which consists of temperature data of electricity transformers; 2) Electricity, which includes power consumption data of several customers; 3) Exchange [24], containing financial exchange rate within a specific time range; 4) Traffic, comprising data related to road traffic; 5) Weather, which involves various weather indicators; and 6) Illness, consisting of recorded influenza-like illness data. Note that ETT is further divided into four sub-datasets: ETTh1, ETTh2, ETTm1, and ETTm2, and the results in Table 1 are based only on the ETTm2 sub-dataset. The results for the remaining three sub-datasets can be found in the Appendix.

**Models for comparison**: In this study, we compare our proposed model against the following state-of-the-art models: four transformer-based models, namely FEDformer [9], Autoformer [8], Pyraformer [13]; one MLP-based model, i.e., Dlinear [4]; and one CNN-based model, i.e., TCN [5]. We also consider two recently proposed models, e.g., N-Hits [2] and FiLM [3]. Due to space

Table 1: Multivariate results for six datasets were obtained using an input length of $I = 96$ (or $I = 36$ for the illness dataset) and output lengths of $O \in \{96, 192, 336, 720\}$ (or $O \in \{24, 36, 48, 60\}$ for the illness dataset). In all experiments, lower MSE values indicate better model performance, and we present the best results in boldface.

| Models | | Fedformer | | Autoformer | | N-HiTS | | Film | | Dlinear | | TCN | | Basisformer | |
|---|---|---|---|---|---|---|---|---|---|---|---|---|---|---|---|
| Metric | | MSE | MAE | MSE | MAE | MSE | MAE | MSE | MAE | MSE | MAE | MSE | MAE | MSE | MAE |
| ETT | 96 | 0.203 | 0.287 | 0.255 | 0.339 | 0.192 | 0.265 | **0.183** | 0.266 | 0.193 | 0.292 | 3.041 | 1.330 | 0.184 | **0.266** |
| | 192 | 0.269 | 0.328 | 0.281 | 0.340 | 0.287 | 0.329 | **0.247** | **0.305** | 0.284 | 0.362 | 3.072 | 1.339 | 0.248 | 0.307 |
| | 336 | 0.325 | 0.366 | 0.339 | 0.372 | 0.389 | 0.389 | **0.309** | **0.343** | 0.369 | 0.554 | 3.105 | 1.348 | 0.321 | 0.355 |
| | 720 | 0.421 | 0.415 | 0.422 | 0.419 | 0.591 | 0.491 | **0.407** | **0.399** | 0.554 | 0.522 | 3.135 | 1.354 | 0.410 | 0.404 |
| electricty | 96 | 0.193 | 0.308 | 0.201 | 0.317 | 1.748 | 1.020 | 0.199 | 0.276 | 0.199 | 0.284 | 0.985 | 0.813 | **0.165** | **0.259** |
| | 192 | 0.201 | 0.315 | 0.222 | 0.334 | 1.743 | 1.018 | 0.198 | 0.279 | 0.198 | 0.287 | 0.996 | 0.821 | **0.178** | **0.272** |
| | 336 | 0.214 | 0.329 | 0.231 | 0.338 | - | - | 0.217 | 0.301 | 0.210 | 0.302 | 1.000 | 0.824 | **0.189** | **0.282** |
| | 720 | 0.246 | 0.355 | 0.254 | 0.361 | - | - | 0.280 | 0.358 | 0.245 | 0.335 | 1.438 | 0.784 | **0.223** | **0.311** |
| exchange | 96 | 0.148 | 0.278 | 0.197 | 0.323 | 1.685 | 1.049 | **0.083** | **0.201** | 0.088 | 0.218 | 3.004 | 1.432 | 0.085 | 0.205 |
| | 192 | 0.271 | 0.380 | 0.300 | 0.369 | 1.658 | 1.023 | 0.179 | 0.300 | 0.176 | 0.315 | 3.048 | 1.444 | 0.177 | **0.299** |
| | 336 | 0.460 | 0.500 | 0.509 | 0.524 | 1.566 | 0.988 | 0.337 | **0.416** | 0.313 | 0.427 | 3.113 | 1.459 | 0.336 | 0.421 |
| | 720 | 1.195 | 0.841 | 1.447 | 0.941 | 1.809 | 1.055 | **0.642** | **0.610** | 0.839 | 0.695 | 3.150 | 1.458 | 0.854 | 0.670 |
| traffic | 96 | 0.587 | 0.366 | 0.613 | 0.388 | 2.138 | 1.026 | 0.652 | 0.395 | 0.650 | 0.396 | 1.438 | 0.784 | **0.444** | **0.315** |
| | 192 | 0.604 | 0.373 | 0.616 | 0.382 | - | - | 0.605 | 0.371 | 0.605 | 0.378 | 1.463 | 0.794 | **0.460** | **0.316** |
| | 336 | 0.621 | 0.383 | 0.622 | 0.387 | - | - | 0.615 | 0.372 | 0.612 | 0.382 | 1.479 | 0.799 | **0.471** | **0.317** |
| | 720 | 0.626 | 0.382 | 0.660 | 0.408 | - | - | 0.692 | 0.428 | 0.645 | 0.394 | 1.499 | 0.804 | **0.486** | **0.318** |
| weather | 96 | 0.217 | 0.296 | 0.266 | 0.336 | 0.648 | 0.492 | 0.193 | 0.234 | 0.196 | 0.255 | 0.615 | 0.589 | **0.173** | **0.214** |
| | 192 | 0.276 | 0.336 | 0.307 | 0.367 | 0.616 | 0.479 | 0.238 | 0.270 | 0.237 | 0.296 | 0.629 | 0.600 | **0.223** | **0.257** |
| | 336 | 0.339 | 0.380 | 0.359 | 0.395 | 0.579 | 0.462 | 0.288 | 0.304 | 0.283 | 0.335 | 0.639 | 0.608 | **0.278** | **0.298** |
| | 720 | 0.403 | 0.428 | 0.419 | 0.428 | 0.541 | 0.447 | 0.358 | 0.350 | 0.343 | 0.383 | 0.639 | 0.610 | **0.355** | **0.347** |
| illness | 24 | 3.228 | 1.260 | 3.486 | 1.287 | 3.297 | 1.679 | 2.198 | 0.911 | 2.398 | 1.040 | 6.624 | 1.830 | **1.550** | **0.814** |
| | 36 | 2.679 | 1.080 | 3.103 | 1.148 | 2.379 | 1.441 | 2.267 | 0.926 | 2.646 | 1.088 | 6.858 | 1.879 | **1.516** | **0.819** |
| | 48 | 2.622 | 1.078 | 2.669 | 1.085 | 3.341 | 1.751 | 2.348 | 0.989 | 2.614 | 1.086 | 6.968 | 1.892 | **1.877** | **0.907** |
| | 60 | 2.857 | 1.157 | 2.770 | 1.125 | 2.278 | 1.493 | 2.508 | 1.038 | 2.804 | 1.146 | 7.127 | 1.918 | **1.878** | **0.902** |

Experiment with '-' means it reported an out-of-memory error on a computer with 80G memory.

constraints, we present the results of a selected number of models in this paper, according to their performance and diversity. Interested readers can refer to the supplementary material for a more comprehensive comparison.

## 4.1 Main results

**Multivariate results**: The results of the multivariate time series forecasting are presented in Table 1. Very frequently, the proposed BasisFormer outperformed the comparison models across all six datasets, achieving the best results. Furthermore, Basisformer exhibited a 21.79% improvement over the state-of-the-art method Fedformer. We also observed that when compared to recently proposed models, such as FiLM [3] and Dlinear [4], BasisFormer significantly improved the average MSE performance by a large margin of 10.78% and 14.78%, respectively. It is worth noting that BasisFormer is vastly superior to the other methods for the traffic dataset, probably because Traffic dataset is a highly periodic dataset and our model can learn periodic representation well.

**Univariate results**: The results of the univariate time series forecasting are presented in Table 2. Our model is on par with the state-of-the-art methods. Specifically, compared to the sota method such as FEDformer [9], our proposed model improved the average MSE performance by 15.36%. Compared to the recent models such as FiLM [3] and Dlinear [4], we also achieved better performance with the increment of 1.6% and 16.17% separately.[3]

## 4.2 Ablation studies

**Effect of learnable basis**: In order to demonstrate the effectiveness of learnable bases, we replaced the learnable basis part in our model with three commonly used types of fixed bases: fixed sine/cosine basis that covers all possible frequencies within the input length, random sine/cosine basis selected in a wide range of frequencies, and covariate embedding which is generally used in Transformer-based models. The results are shown in Table 3. The substitution of the learnable bases resulted in, at minimum, a 5% average decrease in performance. It is pertinent to mention that although

---

[3]Additional experimental details and results are presented in the Appendix section.

Table 2: Univariate results for six datasets were obtained using an input length of $I = 96$ (or $I = 36$ for the illness dataset) and output lengths of $O \in \{96, 192, 336, 720\}$ (or $O \in \{24, 36, 48, 60\}$ for the illness dataset). In all experiments, lower MSE values indicate better model performance, and we present the best results in boldface.

| Models | | Fedformer | | Autoformer | | Informer | | Dlinear | | FiLM | | Basisformer | |
|---|---|---|---|---|---|---|---|---|---|---|---|---|---|
| Metric | | MSE | MAE | MSE | MAE | MSE | MAE | MSE | MAE | MSE | MAE | MSE | MAE |
| ETT | 96 | 0.072 | 0.206 | **0.065** | **0.189** | 0.080 | 0.217 | 0.070 | 0.191 | 0.071 | 0.193 | 0.070 | 0.191 |
| | 192 | 0.102 | 0.245 | 0.118 | 0.256 | 0.112 | 0.259 | 0.104 | 0.238 | 0.101 | **0.236** | **0.101** | 0.238 |
| | 336 | 0.130 | 0.279 | 0.154 | 0.305 | 0.166 | 0.314 | 0.135 | 0.278 | **0.129** | **0.273** | 0.131 | 0.276 |
| | 720 | **0.178** | **0.325** | 0.182 | 0.335 | 0.228 | 0.380 | 0.188 | 0.332 | 0.179 | 0.329 | 0.181 | 0.330 |
| electricty | 96 | **0.253** | 0.370 | 0.341 | 0.438 | 0.258 | **0.367** | 0.374 | 0.439 | 0.400 | 0.453 | 0.333 | 0.408 |
| | 192 | **0.282** | **0.386** | 0.345 | 0.428 | 0.285 | 0.388 | 0.352 | 0.423 | 0.380 | 0.438 | 0.371 | 0.427 |
| | 336 | **0.346** | 0.431 | 0.406 | 0.470 | 0.336 | **0.423** | 0.378 | 0.441 | 0.419 | 0.464 | 0.413 | 0.455 |
| | 720 | **0.422** | **0.484** | 0.565 | 0.581 | 0.607 | 0.599 | 0.419 | 0.479 | 0.472 | 0.506 | 0.471 | 0.498 |
| exchange | 96 | 0.154 | 0.304 | 0.241 | 0.387 | 1.327 | 0.944 | **0.094** | **0.236** | 0.103 | 0.249 | 0.108 | 0.242 |
| | 192 | 0.286 | 0.420 | 0.300 | 0.369 | 1.258 | 0.924 | **0.185** | **0.342** | 0.238 | 0.386 | 0.229 | 0.364 |
| | 336 | 0.511 | 0.555 | 0.509 | 0.524 | 2.179 | 1.296 | **0.340** | **0.458** | 0.355 | 0.489 | 0.436 | 0.499 |
| | 720 | 1.301 | 0.879 | 1.260 | 0.867 | 1.280 | 0.953 | 0.618 | 0.624 | **0.589** | **0.610** | 1.130 | 0.815 |
| traffic | 96 | 0.207 | 0.312 | 0.246 | 0.346 | 0.257 | 0.353 | 0.303 | 0.398 | 0.246 | 0.320 | **0.175** | **0.265** |
| | 192 | 0.205 | 0.312 | 0.266 | 0.370 | 0.299 | 0.376 | 0.256 | 0.345 | 0.208 | 0.284 | **0.169** | **0.259** |
| | 336 | 0.219 | 0.323 | 0.263 | 0.371 | 0.312 | 0.387 | 0.242 | 0.330 | 0.203 | 0.284 | **0.162** | **0.254** |
| | 720 | 0.244 | 0.344 | 0.269 | 0.372 | 0.366 | 0.436 | 0.288 | 0.368 | 0.231 | 0.316 | **0.183** | **0.272** |
| weather | 96 | 0.0062 | 0.062 | 0.0110 | 0.081 | 0.0040 | 0.044 | 0.0050 | 0.057 | **0.0013** | **0.026** | 0.0014 | 0.027 |
| | 192 | 0.0060 | 0.062 | 0.0075 | 0.067 | 0.0020 | 0.040 | 0.0058 | 0.063 | **0.0015** | **0.029** | 0.0016 | 0.030 |
| | 336 | 0.0041 | 0.050 | 0.0063 | 0.062 | 0.0040 | 0.049 | 0.0063 | 0.067 | 0.0017 | 0.030 | **0.0016** | **0.030** |
| | 720 | 0.0055 | 0.059 | 0.0085 | 0.070 | 0.0030 | 0.042 | 0.0064 | 0.066 | 0.0022 | 0.035 | **0.0022** | **0.035** |
| illness | 24 | 0.708 | 0.627 | 0.948 | 0.732 | 5.282 | 2.050 | 0.815 | 0.736 | **0.674** | 0.643 | 0.798 | **0.620** |
| | 36 | 0.584 | 0.717 | 0.634 | 0.650 | 4.554 | 1.916 | 0.914 | 0.817 | 0.758 | 0.731 | **0.563** | **0.605** |
| | 48 | 0.717 | 0.697 | 0.791 | 0.752 | 4.273 | 1.846 | 0.956 | 0.854 | 0.774 | 0.753 | **0.596** | **0.632** |
| | 60 | 0.855 | 0.774 | 0.874 | 0.797 | 5.214 | 2.057 | 1.065 | 0.907 | 0.913 | 0.828 | **0.695** | **0.686** |

Table 3: Comparison of learnable and other basis on the Electricity dataset. The best results are marked in bold.

| basis | | learnable(ours) | | fixed sine/cosine | | random sine/cosine | | covariate embedding | |
|---|---|---|---|---|---|---|---|---|---|
| Metric | | MSE | MAE | MSE | MAE | MSE | MAE | MSE | MAE |
| | 96 | **0.165** | **0.259** | +6.8% | +5.4% | +7.2% | +5.5% | +5.2% | +4.5% |
| | 192 | **0.178** | **0.272** | +4.7% | +3.2% | +6.6% | +4.3% | +3.8% | +3.5% |
| | 336 | **0.189** | **0.282** | +9.2% | +6.1% | +10.1% | +6.5% | +5.2% | +4.6% |
| | 720 | **0.223** | **0.311** | +16.4% | +10.4% | +12.4% | +7.5% | +7.1% | +5.6% |
| | avg | **0.189** | **0.281** | +9.3% | +6.3% | +9.1% | +6.0% | +5.3% | +4.6% |

Table 4: The Impact of multi-head Operations on Basis. In this experiment, k represents the number of heads, with $k \in \{4, 8, 16, 32\}$. We used the Electricity dataset for this experiment. The best results are marked in bold.

| heads | | 4 | | 8 | | 16 | | 32 | |
|---|---|---|---|---|---|---|---|---|---|
| Metric | | MSE | MAE | MSE | MAE | MSE | MAE | MSE | MAE |
| | 96 | 0.177 | 0.273 | 0.168 | 0.263 | **0.165** | **0.259** | 0.170 | 0.264 |
| | 192 | 0.185 | 0.278 | 0.180 | 0.273 | 0.178 | 0.272 | **0.176** | **0.269** |
| | 336 | 0.196 | 0.291 | 0.194 | 0.288 | **0.189** | **0.282** | 0.198 | 0.289 |
| | 720 | 0.248 | 0.331 | **0.221** | 0.311 | 0.223 | **0.311** | 0.225 | 0.312 |
| | avg | 0.201 | 0.293 | 0.191 | 0.284 | 0.189 | 0.281 | 0.199 | 0.283 |

sinusoidal types of bases have good generalization ability, they are inadequate in terms of adaptability for specific data. For covariate embeddings, despite having learnable parameters and containing additional sequence information, they do not account for the distinct correlation between the basis and different time series.

**The Impact of multi-head mechanism:** We further check the impact of the number of heads in the Coef module on performance. The results are displayed in Table 4. The results show that performance presents an upward trend when the number of heads is increased within a specific range. However, beyond a certain number, further augmentation of heads may result in a decline in performance. Consequently, we set $H = 16$ in our experiments.

**Ablation study of the Basis module:** We finally employ ablation experiments to isolate the two different loss functions employed in the Basis module. The results are recorded in Table 5. It can be observed that the InfoNCE loss contributes significantly to the good performance of BasisFormer.

Table 5: The contribution of each loss term in the self-supervised module to performance. The dataset used in this experiment was Electricity. The best results are marked in bold.

| supervision | none | | infonce | | smooth | | infonce+smooth | |
| --- | --- | --- | --- | --- | --- | --- | --- | --- |
| Metric | MSE | MAE | MSE | MAE | MSE | MAE | MSE | MAE |
| 96 | 0.177 | 0.273 | -4.7% | -3.5% | -1.1% | -0.4% | **-6.8%** | **-5.0%** |
| 192 | 0.188 | 0.282 | -3.8% | -3.0% | -0.9% | -0.3% | **-5.5%** | **-3.6%** |
| 336 | 0.205 | 0.298 | -5.5% | -4.0% | -2.8% | -1.6% | **-8.0%** | **-5.3%** |
| 720 | 0.243 | 0.327 | -7.0% | -4.4% | -3.3% | -1.5% | **-8.4%** | **-5.1%** |
| avg | 0.204 | 0.295 | -5.2% | -3.7% | -2.0% | -0.9% | **-7.2%** | **-4.8%** |

Table 6: The impact of the number of bases $N$ on the performance of the model. The electricity dataset is employed in this experiment. We present the best results in boldface.

| basis number | 1 | | 5 | | 10 | | 15 | | 20 | |
| --- | --- | --- | --- | --- | --- | --- | --- | --- | --- | --- |
| Metric | MSE | MAE | MSE | MAE | MSE | MAE | MSE | MAE | MSE | MAE |
| 96 | 0.173 | 0.269 | 0.171 | 0.265 | **0.166** | **0.259** | 0.168 | 0.263 | 0.168 | 0.263 |
| 192 | 0.183 | 0.277 | 0.178 | 0.270 | **0.176** | 0.270 | 0.176 | 0.269 | 0.176 | **0.268** |
| 336 | 0.196 | 0.289 | 0.192 | 0.284 | **0.190** | **0.283** | 0.192 | 0.285 | 0.193 | 0.285 |
| 720 | 0.231 | 0.317 | 0.229 | 0.314 | **0.218** | **0.306** | 0.220 | 0.308 | 0.224 | 0.311 |
| avg | 0.196 | 0.288 | 0.192 | 0.283 | **0.187** | **0.279** | 0.189 | 0.281 | 0.190 | 0.282 |

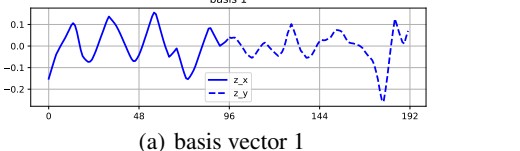

(a) basis vector 1

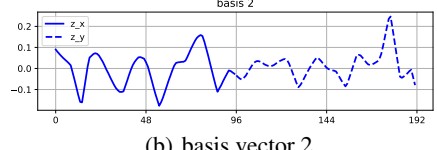

(b) basis vector 2

Figure 2: Two highly correlated basis vectors when the number of basis vectors $N$ is large.

The smoothness loss also makes positive contributions. Remarkably, the employment of both loss functions in conjunction yields the most optimal result. Specifically, the InfoNCE loss advances the bases' capability to acquire the representation of time series, while the smoothness loss function mitigates overfitting to noise within the data.

**Impact of the number of basis vectors**: We present the performance of the proposed model under varying numbers of basis vectors $N$ in Table 6, where $N$ is set to 1, 5, 10, 15, and 20. The results demonstrate that the model's performance remains stable over a wide range of $N$, indicating its ability to adaptively adjust to the number of basis vectors. Notably, when $N$ increases beyond a certain threshold, some of the basis vectors may become redundant. To further explore this, we visualize a subset of the learned basis vectors when $N = 20$ in Figure 2. Interestingly, we observe a high cosine similarity of $-0.93$ between two of the bases, suggesting that some basis vectors may not be necessary for accurate prediction. Thus, in practical applications, we set $N$ to 10 for all datasets to reduce computational complexity without compromising performance.

### 4.3 Other studies

**The adaptability of the self-supervised module to other models:** To demonstrate the generalizability of our self-supervised basis learning module, we treat the covariate used in the FEDformer [9] and Autoformer [8] as learnable bases and supervise them using our self-supervised framework. It is noteworthy that we only provide supervision to the covariate employed in these models, without introducing any additional parameters in the prediction pathway. The results are presented in Table 7. We can tell that the addition of self-supervision to the covariate exclusively causes a performance improvement of approximately 5%. This phenomenon suggests that the learnable basis provides a more reliable reference for the future than the given covariates.

**The interpretability of the bases:** In order to elucidate the impact of basis in a more comprehensible manner, we visualize the time series and the corresponding learned basis for the traffic dataset in Figure 3(a) and Figure 3(b). Specifically, we choose 4 time series and basis vectors at random. Upon observation of the figures, several key points can be discerned: Firstly, the traffic data have strong periodic patterns, with roughly four peak values in a sinusoidal shape within a length of 96. The positions of the peaks are not evenly distributed over time though. Correspondingly, the learned basis

Table 7: The performance comparison of the self-supervised module used in Transformer-based models. The 'origin' represents no modifications made to the original model and the '(+)coef module' represents applying our designed self-supervised network to supervise the covariate in the models. The dataset used in this experiment was Electricity. The best results are marked in bold.

| baseline | Fedformer | | | | Autoformer | | | |
|---|---|---|---|---|---|---|---|---|
| comparison | origin | | (+) coef module | | origin | | (+) coef module | |
| metric | MSE | MAE | MSE | MAE | MSE | MAE | MSE | MAE |
| 96 | 0.193 | 0.308 | **0.183** | **0.301** | 0.201 | 0.317 | **0.193** | **0.305** |
| 192 | 0.201 | 0.315 | **0.196** | **0.312** | 0.222 | 0.334 | **0.200** | **0.312** |
| 336 | 0.214 | 0.329 | **0.209** | **0.324** | 0.231 | 0.338 | **0.211** | **0.325** |
| 720 | 0.246 | 0.355 | **0.229** | **0.340** | 0.254 | 0.361 | **0.252** | **0.359** |
| avg | 0.214 | 0.327 | **0.204** | **0.319** | 0.227 | 0.338 | **0.214** | **0.325** |

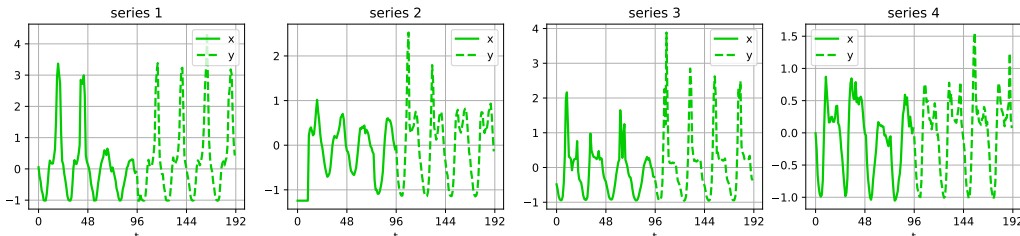

(a) Visualization of time series on the Traffic dataset

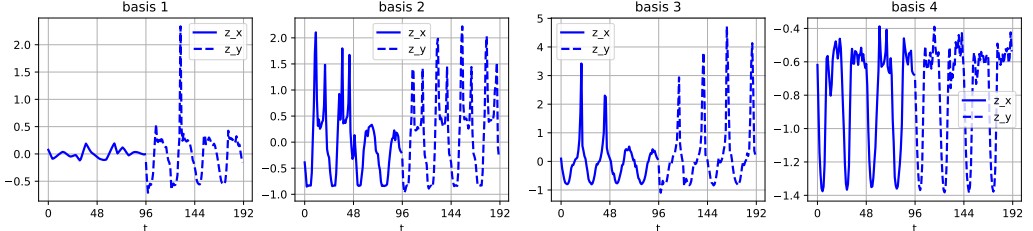

(b) Visualization of learned basis on the Traffic dataset

Figure 3: The visualization of time series and learned basis on the Traffic dataset: The solid line indicates the historical series and the dashed line indicates the future series. For this visualization, we set the input length $I$ to 96 and the output length $O$ to 96.

follows this periodic pattern and effectively capture the salient features of the data. Secondly, the basis vectors given by our approach are smooth, indicating that they are not corrupted by the noise in the data. Note that the noise is not predictable, and so the basis that drives the change in the future is preferred to be smooth. Thirdly, it is evident that the learned bases have varying heights and intervals, thereby providing diversity to characterize different features of the time series. Finally, the obtained bases are consistent from both past and future perspectives, thus facilitating the prediction of future trends based on the coefficient similarity between a time series and the basis in the historical part. More visualized results can be found in the Appendix E.

## 5 Conclusion

This paper presents BasisFormer, a novel solution to alleviate two significant limitations that impede the efficacy of the existing SOTA methods. Through the utilization of BasisFormer, automatic learning of a self-adjusting basis can be achieved. Moreover, given the learned basis, BasisFormer also allows different time series to be correlated with distinct subsets of basis vectors. Our experimental findings provide compelling evidence of the superiority of BasisFormer over existing methods.

## 6 Acknowledgements

The paper is supported in part by the following grants: National Key Research and Development Program of China Grant (No.2018AAA0100400), National Natural Science Foundation of China (No. 62325109, U21B2013, 61971277).

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

# Appendix

## A  Additional Experiments[4]

### A.1  Experiments on the ETT datasets

In the main body, we present a comparison of the benchmark methods on the ETTm2 dataset. In this section, we extend our analysis to the remaining three ETT datasets, namely ETTh1, ETTh2, and ETTm1, as summarized in Table 8. Our experimental results reveal that Basisformer outperforms all other methods in terms of MSE and MAE. Specifically, Basisformer demonstrates a superior average MSE reduction of 1.32% , 6.74% and 9.23% when compared to FiLM, Fedformer and DLinear, respectively.

Table 8: Multivariate results for the remaining three ETT datasets using an input length of $I = 96$ (or $I = 36$ for the illness dataset) and output lengths of $O \in \{96, 192, 336, 720\}$ (or $O \in \{24, 36, 48, 60\}$ for the illness dataset). In all experiments, lower MSE values indicate better model performance, and we present the best results in boldface.

| Models | | Fedformer | | Autoformer | | N-HiTS | | FiLM | | Dlinear | | Informer | | Basisformer | |
|---|---|---|---|---|---|---|---|---|---|---|---|---|---|---|---|---|
| Metric | | MSE | MAE | MSE | MAE | MSE | MAE | MSE | MAE | MSE | MAE | MSE | MAE | MSE | MAE |
| ETTh1 | 96 | **0.376** | 0.419 | 0.449 | 0.459 | 0.419 | 0.413 | 0.388 | 0.401 | 0.386 | **0.400** | 0.865 | 0.713 | 0.394 | 0.411 |
| | 192 | **0.420** | 0.448 | 0.500 | 0.482 | 0.468 | 0.443 | 0.443 | 0.439 | 0.437 | **0.432** | 1.008 | 0.792 | 0.442 | 0.437 |
| | 336 | **0.459** | 0.465 | 0.521 | 0.496 | 0.551 | 0.489 | 0.484 | 0.461 | 0.481 | 0.459 | 1.107 | 0.809 | 0.473 | **0.451** |
| | 720 | 0.506 | 0.507 | 0.514 | 0.512 | 0.669 | 0.559 | 0.525 | 0.519 | 0.519 | 0.516 | 1.181 | 0.865 | **0.460** | **0.465** |
| ETTh2 | 96 | 0.358 | 0.397 | 0.346 | 0.388 | 0.374 | 0.383 | **0.292** | **0.341** | 0.333 | 0.387 | 3.755 | 1.525 | 0.312 | 0.356 |
| | 192 | 0.429 | 0.439 | 0.456 | 0.452 | 0.476 | 0.446 | **0.378** | **0.396** | 0.477 | 0.476 | 5.602 | 1.931 | 0.382 | 0.401 |
| | 336 | 0.496 | 0.487 | 0.482 | 0.486 | 0.472 | 0.446 | 0.426 | 0.438 | 0.594 | 0.541 | 4.721 | 1.835 | **0.418** | **0.431** |
| | 720 | 0.463 | 0.474 | 0.515 | 0.511 | 0.932 | 0.636 | 0.443 | 0.455 | 0.831 | 0.657 | 3.647 | 1.625 | **0.418** | **0.438** |
| ETTm1 | 96 | 0.379 | 0.419 | 0.505 | 0.475 | **0.324** | **0.349** | 0.357 | 0.373 | 0.345 | 0.372 | 0.672 | 0.571 | 0.342 | 0.374 |
| | 192 | 0.426 | 0.441 | 0.553 | 0.496 | **0.376** | **0.379** | 0.387 | 0.385 | 0.380 | 0.389 | 0.795 | 0.669 | 0.380 | 0.392 |
| | 336 | 0.445 | 0.459 | 0.621 | 0.537 | **0.409** | **0.405** | 0.420 | 0.407 | 0.413 | 0.413 | 1.212 | 0.871 | 0.420 | 0.418 |
| | 720 | 0.543 | 0.490 | 0.671 | 0.561 | **0.472** | 0.443 | 0.478 | **0.439** | 0.474 | 0.453 | 1.166 | 0.823 | 0.492 | 0.458 |

### A.2  Experimental results with longer length input setting

Throughout our research, we maintain consistency in our experimental settings by fixing the input length to be 96 (with a reduced input length of 36 for the illness dataset), instead of using a longer length. The main rationale behind this decision is that, in practical scenarios where the model is deployed as an online service and tasked with predicting a long range of the future at a granular level of minutes or hours, collecting a lengthy history (i.e., spanning 720 timestamps) for a large number of time series in real-time can be quite challenging. Therefore, the adoption of an input length of 96 proves to be more practical and feasible.

Given that certain recent methods utilize longer input lengths to yield better performance, irrespective of the length, we present supplementary comparison outcomes with extended input lengths in Table 9. Specifically, Fedformer, Autoformer, and TCN exhibit a decline in performance with an increase in input length, and hence, we retain their original outcomes at an input length of 96. In contrast, Dlinear employs an input length of 336 (104 for the illness dataset) by default, FiLM utilizes an input length that is at most four times of the output length, and N-HiTS adopts an input length that is five times of the output length. To enable a fair comparison, we standardize our input length for longer inputs to 192 (72 for the illness dataset).

The experimental results yield several notable findings. Firstly, those methods that benefit from longer inputs, namely Dlinear, FiLM, and N-HiTS, exhibit a significant performance decline when the input length is reduced from longer settings to an input length of 96. Concretely, Dlinear, FiLM, and N-HiTS show performance declines of 25.82%, 19.48%, and 330.42%, respectively. Conversely, our approach maintains most of its performance with a slight deterioration of 6.23%, as evident in Table 1 and Table 9. Secondly, concerning longer inputs, our method surpasses recent approaches such as Dlinear, FiLM, and N-HiTS, with an average MSE performance improvement of 1.35%, 0.63%, and 7.75%, respectively, and a corresponding evaluation MAE performance improvement of 3.15%, 2.33%, and 4.06%, respectively. It is noteworthy that our approach requires an input length

---

[4]All the six datasets can be downloaded from `https://drive.google.com/drive/folders/1ZOYpTUa82_jCcxIdTmyr0LXQfvaM9vIy?usp=sharing`

Table 9: Multivariate results for six datasets using a longer input length. Lower MSE indicate superior model performance, and the best results are presented in boldface.

| Models | | Fedformer | | Autoformer | | N-HiTS | | FiLM | | Dlinear | | TCN | | Basisformer | |
|---|---|---|---|---|---|---|---|---|---|---|---|---|---|---|---|
| Metric | | MSE | MAE | MSE | MAE | MSE | MAE | MSE | MAE | MSE | MAE | MSE | MAE | MSE | MAE |
| ETT | 96 | 0.203 | 0.287 | 0.255 | 0.339 | 0.176 | **0.255** | **0.165** | 0.256 | 0.167 | 0.260 | 3.041 | 1.330 | 0.185 | 0.270 |
| | 192 | 0.269 | 0.328 | 0.281 | 0.340 | 0.245 | 0.305 | **0.222** | **0.296** | 0.224 | 0.303 | 3.072 | 1.339 | 0.247 | 0.307 |
| | 336 | 0.325 | 0.366 | 0.339 | 0.372 | 0.295 | 0.346 | **0.277** | **0.333** | 0.281 | 0.342 | 3.105 | 1.348 | 0.298 | 0.341 |
| | 720 | 0.421 | 0.415 | 0.422 | 0.419 | 0.401 | 0.416 | **0.371** | **0.389** | 0.397 | 0.421 | 3.135 | 1.354 | 0.381 | 0.393 |
| electricty | 96 | 0.193 | 0.308 | 0.201 | 0.317 | 0.147 | 0.249 | 0.154 | 0.267 | **0.140** | **0.237** | 0.985 | 0.813 | 0.145 | 0.245 |
| | 192 | 0.201 | 0.315 | 0.222 | 0.334 | 0.167 | 0.269 | 0.164 | 0.258 | **0.153** | **0.249** | 0.996 | 0.821 | 0.165 | 0.263 |
| | 336 | 0.214 | 0.329 | 0.231 | 0.338 | 0.186 | 0.290 | 0.188 | 0.283 | **0.169** | **0.267** | 1.000 | 0.824 | 0.178 | 0.276 |
| | 720 | 0.246 | 0.355 | 0.254 | 0.361 | 0.243 | 0.340 | 0.236 | 0.332 | **0.203** | **0.301** | 1.438 | 0.784 | 0.219 | 0.310 |
| exchange | 96 | 0.148 | 0.278 | 0.197 | 0.323 | 0.092 | 0.211 | **0.079** | 0.204 | 0.081 | **0.203** | 3.004 | 1.432 | 0.084 | 0.205 |
| | 192 | 0.271 | 0.380 | 0.300 | 0.369 | 0.208 | 0.322 | 0.159 | **0.292** | **0.157** | 0.293 | 3.048 | 1.444 | 0.172 | 0.298 |
| | 336 | 0.460 | 0.500 | 0.509 | 0.524 | 0.341 | 0.422 | **0.270** | **0.398** | 0.305 | 0.414 | 3.113 | 1.459 | 0.303 | 0.403 |
| | 720 | 1.195 | 0.841 | 1.447 | 0.941 | 0.888 | 0.723 | **0.536** | **0.574** | 0.643 | 0.601 | 3.150 | 1.458 | 0.781 | 0.668 |
| traffic | 96 | 0.587 | 0.366 | 0.613 | 0.388 | **0.402** | **0.282** | 0.416 | 0.294 | 0.410 | 0.282 | 1.438 | 0.784 | 0.403 | 0.293 |
| | 192 | 0.604 | 0.373 | 0.616 | 0.382 | 0.420 | 0.297 | **0.408** | **0.288** | 0.423 | 0.287 | 1.463 | 0.794 | 0.421 | 0.301 |
| | 336 | 0.621 | 0.383 | 0.622 | 0.387 | 0.448 | 0.313 | 0.425 | 0.298 | 0.436 | **0.296** | 1.479 | 0.799 | **0.418** | 0.298 |
| | 720 | 0.626 | 0.382 | 0.660 | 0.408 | 0.539 | 0.353 | 0.520 | 0.353 | 0.466 | 0.315 | 1.499 | 0.804 | **0.464** | **0.312** |
| weather | 96 | 0.217 | 0.296 | 0.266 | 0.336 | **0.158** | **0.195** | 0.199 | 0.262 | 0.176 | 0.237 | 0.615 | 0.589 | 0.168 | 0.215 |
| | 192 | 0.276 | 0.336 | 0.307 | 0.367 | **0.211** | **0.247** | 0.228 | 0.288 | 0.220 | 0.282 | 0.629 | 0.600 | 0.213 | 0.257 |
| | 336 | 0.339 | 0.380 | 0.359 | 0.395 | 0.274 | 0.300 | 0.267 | 0.323 | 0.265 | 0.319 | 0.639 | 0.608 | **0.263** | **0.292** |
| | 720 | 0.403 | 0.428 | 0.419 | 0.428 | 0.351 | 0.353 | **0.319** | 0.361 | 0.323 | 0.362 | 0.639 | 0.610 | 0.343 | **0.346** |
| illness | 24 | 3.228 | 1.260 | 3.486 | 1.287 | 1.862 | 0.869 | 1.970 | 0.875 | 2.215 | 1.081 | 6.624 | 1.830 | **1.427** | **0.778** |
| | 36 | 2.679 | 1.080 | 3.103 | 1.148 | 2.071 | 0.969 | 1.982 | 0.859 | 1.936 | 0.963 | 6.858 | 1.879 | **1.464** | **0.813** |
| | 48 | 2.622 | 1.078 | 2.669 | 1.085 | 2.184 | 0.999 | 1.868 | 0.896 | 2.130 | 1.024 | 6.968 | 1.892 | **1.660** | **0.862** |
| | 60 | 2.857 | 1.157 | 2.770 | 1.125 | 2.507 | 1.060 | 2.057 | 0.929 | 2.368 | 1.096 | 7.127 | 1.918 | **1.853** | **0.917** |

of 192 (72 for the illness dataset), which is at least 40% lower than the input length of the other three methods. Furthermore, for even longer input lengths, our model's performance can be further enhanced, signifying that our approach can leverage limited data more efficiently.

We simultaneously study the inference time of our model for various input lengths, denoted as $I \in \{96, 192, 336, 720\}$, and output lengths, denoted as $O \in \{96, 192, 336, 720\}$. The results are shown in Table 10. These measurements are based on the "exchange" dataset.

Table 10: Inference time for various input/output settings

| | O=96 | O=192 | O=336 | O=720 |
|---|---|---|---|---|
| I=96 | 0.000833 | 0.001211 | 0.001419 | 0.00211 |
| I=192 | 0.000884 | 0.001285 | 0.001437 | 0.002139 |
| I=336 | 0.000893 | 0.001338 | 0.001469 | 0.002194 |
| I=720 | 0.000941 | 0.001364 | 0.001547 | 0.002246 |

The table above illustrates the average inference time per instance of our algorithm under different configurations, measured in seconds. As observed, our algorithm exhibits notable speed, averaging at the millisecond level. Furthermore, when comparing the increase in output length to the extension of the input length, the additional inference time incurred by augmenting the input length is minimal. This is attributed to the preprocessing step where the input sequence, regardless of its length, is projected into a fixed-length (usually 100) sequence using a linear layer. As a result, extending the input length does not significantly amplify the time consumption in our method.

## A.3   Additional abalation study

**Impact of the number of the BCAB layers**: The ablation study on the number of BCABs is shown in Table 11. The findings indicate that stacking a certain number of BCAB modules can enhance the performance of the model. However, exceeding a certain threshold can lead to overfitting, resulting in a decline in performance. Hence, we recommend the use of two layers of BCABs in practical experiments to achieve optimal performance without overfitting the model.

**Impact of the bottleneck in the forecast module:** The performance of the proposed model under varying bottleneck settings is presented in Table 12. The results demonstrate that employing a

Table 11: The impact of the number of stacked BCAB on the performance of the model. The electricity dataset is employed in this experiment. We present the best results in boldface.

| BCAB number | 0 | | 1 | | 2 | | 3 | | 4 | |
|---|---|---|---|---|---|---|---|---|---|---|
| Metric | MSE | MAE | MSE | MAE | MSE | MAE | MSE | MAE | MSE | MAE |
| 96 | 0.186 | 0.273 | 0.166 | 0.260 | **0.166** | **0.259** | 0.168 | 0.263 | 0.171 | 0.266 |
| 192 | 0.187 | 0.274 | 0.176 | 0.270 | **0.176** | **0.268** | 0.176 | 0.269 | 0.179 | 0.272 |
| 336 | 0.208 | 0.297 | **0.187** | **0.280** | 0.190 | 0.283 | 0.190 | 0.283 | 0.191 | 0.284 |
| 720 | 0.244 | 0.325 | 0.228 | 0.313 | **0.218** | **0.306** | 0.234 | 0.319 | 0.237 | 0.319 |
| avg | 0.206 | 0.292 | 0.189 | 0.281 | **0.187** | **0.279** | 0.192 | 0.283 | 0.194 | 0.285 |

bottleneck architecture with a width of 48 can significantly reduce the number of model parameters without degrading the performance significantly, as opposed to not using a bottleneck architecture.

Table 12: The impact of the MLP bottleneck in the forecast module. The electricity dataset is employed in this experiment. Setting the bottleneck dimension to 96 is equivalent to not using a bottleneck since the input length is 96. The best results are highlighted in bold. The second best is underlined.

| bottleneck | 96 | | 48 | | 32 | | 24 | |
|---|---|---|---|---|---|---|---|---|
| Metric | MSE | MAE | MSE | MAE | MSE | MAE | MSE | MAE |
| 96 | **0.163** | **0.257** | 0.166 | 0.259 | 0.172 | 0.267 | 0.172 | 0.269 |
| 192 | **0.172** | **0.265** | 0.176 | 0.268 | 0.182 | 0.273 | 0.186 | 0.279 |
| 336 | **0.186** | **0.279** | 0.190 | 0.283 | 0.194 | 0.286 | 0.197 | 0.289 |
| 720 | **0.217** | **0.305** | 0.218 | 0.306 | 0.230 | 0.316 | 0.233 | 0.317 |
| avg | **0.184** | **0.276** | 0.187 | 0.279 | 0.195 | 0.286 | 0.197 | 0.288 |

## A.4 Sensitivity analysis of the weights for the losses in Eq.(9)

Our model utilizes three distinct loss functions: the supervised MSE loss for prediction $L_{pred}$, the self-supervised InfoNCE loss for basis learning $L_{align}$, and the smoothness loss for smoothing the basis over time $L_{align}$. During training, we directly combine these loss functions as the model's performance is not significantly impacted by the relative weights of the individual losses within a certain range. This assertion is supported by the performance evaluation presented in Figure 4, which investigates the impact of different weight combinations of the three loss functions. In our setting, we fix the weight of the predicted loss function to be 1, and then fix either the weight of the contrast loss function or the smoothness loss function to be 1, while the other one varies within the range of $\{0.2, 0.4, 0.6, 0.8, 1.0, 1.2, 1.4, 1.6\}$. To explore the inflection point of the effect, we take the middle point of two points and calculate a finer range again. Our results indicate that the contrast loss function is essentially stable between the weight range of 0.6-1.2, while the smoothness loss function is similarly stable between the weight range of 0.9-1.5.

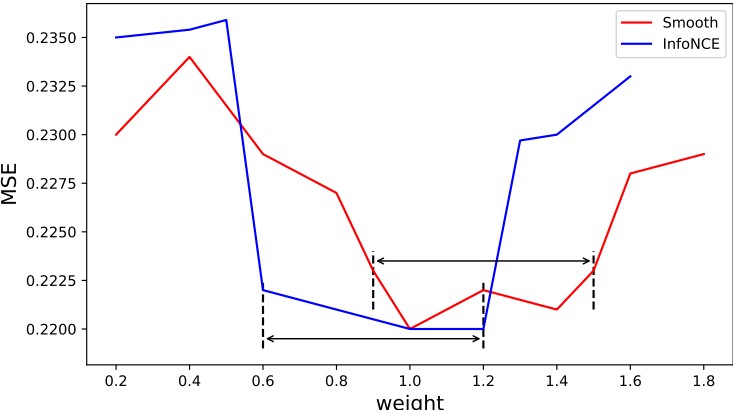

Figure 4: MSE for the testing data as a function of the weight for the smoothness (the red line) and the infoNCE loss(the blue line).

## A.5 Comparison with self-supervised methods

Our proposed method adopts contrastive learning, here we provide comparative experiments with other self-supervised learning method namely Cost [25]. For the experiments, we fix the input length at 96 and vary the output length from 96 to 720. Table 13 and Table 14 illustrate that Basisformer outperforms Cost in terms of both MSE and MAE across all cases. It is important to note that Cost is trained using a two-step process: self-supervised training followed by supervised ridge regression. In contrast, Basisformer trains the basis, coef, and forecast modules end-to-end. This enables Basisformer to learn a basis that is specifically tailored for forecasting, potentially leading to improved performance.

Table 13: Multidimensional prediction comparative experiments on three datasets.

| Models | Metric | ETT | | | | Electricity | | | | exchange | | | |
| --- | --- | --- | --- | --- | --- | --- | --- | --- | --- | --- | --- | --- | --- |
| | | 96 | 192 | 336 | 720 | 96 | 192 | 336 | 720 | 96 | 192 | 336 | 720 |
| Basisformer | MSE | **0.184** | **0.248** | **0.321** | **0.410** | **0.165** | **0.178** | **0.189** | **0.223** | **0.085** | **0.177** | **0.336** | **0.854** |
| | MAE | **0.266** | **0.307** | **0.355** | **0.404** | **0.259** | **0.272** | **0.282** | **0.311** | **0.205** | **0.299** | **0.421** | **0.670** |
| Cost | MSE | 0.280 | 0.480 | 0.805 | 1.562 | 0.199 | 0.199 | 0.212 | 0.246 | 0.263 | 0.464 | 0.833 | 1.192 |
| | MAE | 0.375 | 0.506 | 0.676 | 0.955 | 0.290 | 0.292 | 0.307 | 0.338 | 0.393 | 0.521 | 0.691 | 0.871 |

Table 14: Multidimensional prediction comparative experiments on another three datasets.

| Models | Metric | traffic | | | | weather | | | | illness | | | |
| --- | --- | --- | --- | --- | --- | --- | --- | --- | --- | --- | --- | --- | --- |
| | | 96 | 192 | 336 | 720 | 96 | 192 | 336 | 720 | 96 | 192 | 336 | 720 |
| Basisformer | MSE | **0.444** | **0.460** | **0.471** | **0.486** | **0.173** | **0.223** | **0.278** | **0.355** | **1.550** | **1.516** | **1.877** | **1.878** |
| | MAE | **0.315** | **0.316** | **0.317** | **0.318** | **0.214** | **0.257** | **0.298** | **0.347** | **0.814** | **0.819** | **0.907** | **0.902** |
| Cost | MSE | 0.576 | 0.546 | 0.555 | 0.591 | 0.372 | 0.528 | 0.835 | 1.394 | 2.330 | 2.497 | 2.650 | 2.829 |
| | MAE | 0.377 | 0.359 | 0.363 | 0.379 | 0.415 | 0.517 | 0.666 | 0.894 | 0.923 | 0.962 | 1.032 | 1.062 |

## A.6 Comparison with time series discretization

To compare with time series discretization, we have chosen an method, namely Boss [26], which utilizes the Bag-Of-SFA-symbols method for feature extraction.

In order to evaluate the effectiveness of our approach, we have conducted a classification task using several UCR datasets. The description of the datasets is concluded in Table 15:

Table 15: The description of selected UCR datasets

| Dataset | Train Size | Test Size | Length | Classes | Is_predictable | Description |
| --- | --- | --- | --- | --- | --- | --- |
| Mallat | 55 | 2345 | 1024 | 8 | Y | a simulated dataset |
| Rock | 20 | 50 | 2844 | 4 | Y | rock examples from the ASTER spectral library |
| Phoneme | 214 | 1896 | 1024 | 39 | N | Each series is extracted from the segmented audio |
| FaceUCR | 200 | 2050 | 131 | 14 | N | rotationally aligned version of facial outline |

To adapt our method for classification, we followed these steps:

We partitioned the sequence into past and future parts, uniformly dividing them in a 6:4 ratio for all datasets. Different partitioning methods can be explored in future research to improve the model's performance.

We used a self-supervised approach for training, reserving 10% of the original training data for validating self-supervised performance. The remaining data was used for training, and the self-supervised loss function included prediction, alignment, and smoothness losses. Early termination based on validation set performance was done with a patience of 3.

From the well-trained self-supervised model, we extracted the aggregation coefficient matrix, specifically from the past perspective. This matrix was flattened to create sequence features, which were then fed into a random forest classifier for final classification. Notably, during self-supervised training, both past and future sequences were used for consistency, but only the past coefficient matrix was utilized in the classifier.

We conducted a fair comparison by extracting features using both Boss and our model, ensuring that our feature parameter count did not exceed Boss's. We employed a random forest classifier with 100 features and a maximum depth of 30 for classification, and the results are summarized in the Table 16.

Table 16: The comparison results of BOSS and Basisformer. We present the best results in boldface.

| Model | Boss | Basisformer | |
|---|---|---|---|
| Metric | acc | acc | valid_loss |
| Mallat | 0.83 | **0.87** | 0.12 |
| Rock | 0.56 | **0.72** | 0.18 |
| Phoneme | **0.20** | 0.07 | 1.27 |
| FaceUCR | **0.68** | 0.41 | 1.68 |

The applicability of our method to self-supervision relies on predictability and consistency between past and future data. The validation set loss in the table indicates that datasets lacking predictability have high validation losses, posing challenges for loss function optimization. The Phoneme and FaceUCR datasets lack predictability. The Phoneme dataset includes speech segments from different individuals with random content before and after, while the FaceUCR dataset consists of flattened one-dimensional vectors of rotated face images, both lacking inherent predictability. These datasets require a holistic understanding of the entire sequence for meaningful interpretation, and as a result, our proposed Basisformer struggles to extract useful features, leading to lower performance compared to Boss.

On the other hand, datasets like Mallat and Rock, exhibiting predictability and low validation losses, allow our approach to achieve superior performance over Boss. Surprisingly, we achieve this performance using only the representation of the past sequence as input for the classifier.

### A.7 Uncertainty of the results

To assess the stability of our proposed method, we performed 5 repeated experiments and calculated the standard deviations for all methods, as presented in Table 17. Notably, our method exhibits a relatively small variance within the table, indicating its high degree of stability.

## B Implementation Details

The training and testing of BasisFormer are conducted on an NVIDIA GeForce RTX 3090 graphics card with 24268MB of VRAM. During the trainin process, we adopt the Adabelief optimizer [27] for optimization. We train the model for 30 epochs with the patience of 3 epochs. All experiments are averaged over 5 trials.

To implement the multi-head mechanism, we calculate the multi-head attention for each CAB separately, and then restore it to the original dimension through multiplication, concatenation, and a linear layer. In the last layer of the network, a mapping layer was utilized to map it to $H$ heads, and the dot product outputs the final coefficients.

To promote the learning of bases and ensure consistency of time series across different dimensions, we normalized the time series during training and performed inverse normalization when outputting the results.

For the other models compared in the table, we utilized their original code and conducted experiments by only varying the input length.

## C Analysis of the Limitations of BasisFormer

BasisFormer demonstrates proficiency in learning effective representations and capturing the relationship between bases and time series. However, this proficiency is contingent upon the multi-dimensional time series being on the same feature scale, which necessitates normalization of the time series during training and inverse normalization when outputting results. Despite this, the normalization and inverse normalization operations introduce changes to the original distribution of

Table 17: Results for 6 benchmark datasets with standard deviations in the brackets.

| Models | | Fedformer | | Autoformer | | N-HiTS | | FiLM | | Dlinear | | TCN | | Basisformer | |
|---|---|---|---|---|---|---|---|---|---|---|---|---|---|---|---|
| Metric | | MSE | MAE | MSE | MAE | MSE | MAE | MSE | MAE | MSE | MAE | MSE | MAE | MSE | MAE |
| ETT | 96 | 0.203 (0.002) | 0.287 (0.001) | 0.255 (0.020) | 0.339 (0.020) | 0.192 (0.003) | 0.265 (0.002) | **0.183** (0.000) | 0.266 (0.000) | 0.193 (0.004) | 0.292 (0.006) | 3.041 (0.000) | 1.330 (0.000) | 0.184 (0.002) | **0.266** (0.002) |
| | 192 | 0.269 (0.006) | 0.328 (0.005) | 0.281 (0.027) | 0.340 (0.025) | 0.287 (0.004) | 0.329 (0.001) | **0.247** (0.000) | **0.305** (0.000) | 0.284 (0.016) | 0.362 (0.016) | 3.072 (0.002) | 1.339 (0.001) | 0.248 (0.004) | 0.307 (0.002) |
| | 336 | 0.325 (0.002) | 0.366 (0.003) | 0.339 (0.018) | 0.372 (0.015) | 0.389 (0.005) | 0.389 (0.003) | **0.309** (0.000) | **0.343** (0.000) | 0.369 (0.006) | 0.554 (0.002) | 3.105 (0.005) | 1.348 (0.003) | 0.321 (0.005) | 0.355 (0.004) |
| | 720 | 0.421 (0.018) | 0.415 (0.012) | 0.422 (0.015) | 0.419 (0.010) | 0.591 (0.011) | 0.491 (0.002) | **0.407** (0.001) | **0.399** (0.000) | 0.554 (0.037) | 0.522 (0.026) | 3.135 (0.021) | 1.354 (0.005) | 0.410 (0.007) | 0.404 (0.004) |
| electricity | 96 | 0.193 (0.001) | 0.308 (0.001) | 0.201 (0.003) | 0.317 (0.004) | 1.748 (0.003) | 1.020 (0.001) | 0.199 (0.000) | 0.276 (0.001) | 0.199 (0.000) | 0.284 (0.000) | 0.985 (0.006) | 0.813 (0.004) | **0.165** (0.001) | **0.259** (0.001) |
| | 192 | 0.201 (0.005) | 0.315 (0.006) | 0.222 (0.003) | 0.334 (0.004) | 1.743 (0.008) | 1.018 (0.003) | 0.198 (0.000) | 0.279 (0.001) | 0.198 (0.000) | 0.287 (0.000) | 0.996 (0.008) | 0.821 (0.007) | **0.178** (0.001) | **0.272** (0.001) |
| | 336 | 0.214 (0.001) | 0.329 (0.002) | 0.231 (0.006) | 0.338 (0.004) | 1.677 (0.010) | 1.000 (0.003) | 0.217 (0.001) | 0.301 (0.001) | 0.210 (0.001) | 0.302 (0.001) | 1.000 (0.004) | 0.824 (0.003) | **0.189** (0.001) | **0.282** (0.001) |
| | 720 | 0.246 (0.003) | 0.355 (0.003) | 0.254 (0.007) | 0.361 (0.008) | - | - | 0.280 (0.000) | 0.358 (0.000) | 0.245 (0.000) | 0.335 (0.000) | 1.438 (0.006) | 0.784 (0.003) | **0.223** (0.002) | **0.311** (0.001) |
| exchange | 96 | 0.148 (0.004) | 0.278 (0.004) | 0.197 (0.019) | 0.323 (0.012) | 1.685 (0.042) | 1.049 (0.017) | **0.083** (0.003) | **0.201** (0.003) | 0.088 (0.004) | 0.218 (0.005) | 3.004 (0.128) | 1.432 (0.070) | 0.085 (0.004) | 0.205 (0.005) |
| | 192 | 0.271 (0.012) | 0.380 (0.010) | 0.300 (0.020) | 0.369 (0.016) | 1.658 (0.015) | 1.023 (0.006) | 0.179 (0.003) | 0.300 (0.002) | **0.176** (0.005) | 0.315 (0.006) | 3.048 (0.020) | 1.444 (0.008) | 0.177 (0.005) | **0.299** (0.005) |
| | 336 | 0.460 (0.009) | 0.500 (0.007) | 0.509 (0.041) | 0.524 (0.016) | 1.566 (0.037) | 0.988 (0.015) | 0.337 (0.005) | **0.416** (0.003) | **0.313** (0.008) | 0.427 (0.006) | 3.113 (0.082) | 1.459 (0.021) | 0.336 (0.011) | 0.421 (0.007) |
| | 720 | 1.195 (0.042) | 0.841 (0.017) | 1.447 (0.084) | 0.941 (0.028) | 1.809 (0.052) | 1.055 (0.018) | **0.642** (0.040) | **0.610** (0.029) | 0.839 (0.027) | 0.695 (0.012) | 3.150 (0.237) | 1.458 (0.063) | 0.854 (0.024) | 0.670 (0.011) |
| traffic | 96 | 0.587 (0.010) | 0.366 (0.008) | 0.613 (0.028) | 0.388 (0.012) | 2.138 (0.016) | 1.026 (0.006) | 0.652 (0.001) | 0.395 (0.003) | 0.650 (0.001) | 0.396 (0.001) | 1.438 (0.001) | 0.784 (0.001) | **0.444** (0.003) | **0.315** (0.003) |
| | 192 | 0.604 (0.012) | 0.373 (0.009) | 0.616 (0.042) | 0.382 (0.020) | 2.101 (0.015) | 1.015 (0.007) | 0.605 (0.001) | 0.371 (0.003) | 0.605 (0.002) | 0.378 (0.001) | 1.463 (0.032) | 0.794 (0.010) | **0.460** (0.004) | **0.316** (0.002) |
| | 336 | 0.621 (0.008) | 0.383 (0.008) | 0.622 (0.009) | 0.387 (0.003) | - | - | 0.615 (0.001) | 0.372 (0.001) | 0.612 (0.003) | 0.382 (0.004) | 1.479 (0.003) | 0.799 (0.002) | **0.471** (0.005) | **0.317** (0.004) |
| | 720 | 0.626 (0.004) | 0.382 (0.003) | 0.660 (0.025) | 0.408 (0.015) | - | - | 0.692 (0.000) | 0.428 (0.000) | 0.645 (0.001) | 0.394 (0.001) | 1.499 (0.010) | 0.804 (0.005) | **0.486** (0.005) | **0.318** (0.004) |
| weather | 96 | 0.217 (0.018) | 0.296 (0.019) | 0.266 (0.007) | 0.336 (0.006) | 0.648 (0.001) | 0.492 (0.000) | 0.193 (0.002) | 0.234 (0.001) | 0.196 (0.001) | 0.255 (0.003) | 0.615 (0.002) | 0.589 (0.002) | **0.173** (0.003) | **0.214** (0.003) |
| | 192 | 0.276 (0.015) | 0.336 (0.017) | 0.307 (0.024) | 0.367 (0.022) | 0.616 (0.003) | 0.479 (0.001) | 0.238 (0.000) | 0.270 (0.001) | 0.237 (0.001) | 0.296 (0.002) | 0.629 (0.023) | 0.600 (0.009) | **0.223** (0.002) | **0.257** (0.001) |
| | 336 | 0.339 (0.014) | 0.380 (0.015) | 0.359 (0.035) | 0.395 (0.031) | 0.579 (0.002) | 0.462 (0.001) | 0.288 (0.001) | 0.304 (0.000) | 0.283 (0.002) | 0.335 (0.004) | 0.639 (0.050) | 0.608 (0.017) | **0.278** (0.001) | **0.298** (0.000) |
| | 720 | 0.403 (0.009) | 0.428 (0.008) | 0.419 (0.017) | 0.428 (0.014) | 0.541 (0.001) | 0.447 (0.000) | 0.358 (0.001) | 0.350 (0.000) | 0.343 (0.020) | 0.383 (0.020) | 0.639 (0.050) | 0.610 (0.018) | **0.355** (0.001) | **0.347** (0.001) |
| illness | 24 | 3.228 (0.020) | 1.260 (0.009) | 3.486 (0.107) | 1.287 (0.018) | 3.297 (0.007) | 1.679 (0.000) | 2.198 (0.138) | 0.911 (0.058) | 2.398 (0.065) | 1.040 (0.032) | 6.624 (0.550) | 1.830 (0.094) | **1.550** (0.087) | **0.814** (0.024) |
| | 36 | 2.679 (0.018) | 1.080 (0.005) | 3.103 (0.139) | 1.148 (0.025) | 2.379 (0.136) | 1.441 (0.043) | 2.267 (0.077) | 0.926 (0.059) | 2.646 (0.137) | 1.088 (0.064) | 6.858 (0.216) | 1.879 (0.034) | **1.516** (0.130) | **0.819** (0.030) |
| | 48 | 2.622 (0.010) | 1.078 (0.002) | 2.669 (0.151) | 1.085 (0.037) | 3.341 (0.092) | 1.751 (0.030) | 2.348 (0.115) | 0.989 (0.037) | 2.614 (0.140) | 1.086 (0.049) | 6.968 (0.032) | 1.892 (0.008) | **1.877** (0.110) | **0.907** (0.032) |
| | 60 | 2.857 (0.011) | 1.157 (0.003) | 2.770 (0.085) | 1.125 (0.019) | 2.278 (0.187) | 1.493 (0.064) | 2.508 (0.130) | 1.038 (0.018) | 2.804 (0.049) | 1.146 (0.009) | 7.127 (0.134) | 1.918 (0.025) | **1.878** (0.098) | **0.902** (0.024) |

Experiment with '-' means it reported an out-of-memory error on a computer with 128G memory.

the time series, making it arduous to fit certain distributions. As such, future work could explore alternative approaches to training on datasets with considerably different feature scales, eliminating the need for normalization and inverse normalization. Possible avenues for investigation include identifying appropriate mathematical methods or neural network transformations to map data to a suitable and universal feature space.

## D   Relation to Meta-learning

From a meta-learning standpoint, the learnable basis in our model is tantamount to meta-knowledge for all time series within the same window. The coefficients, which are derived from the similarity between each time series and the foundation, represent distinctive knowledge for each time series. Consequently, our model can be perceived as a manifestation of meta-learning. Notwithstanding, we departed from conventional meta-learning approaches by forgoing a two-stage inner-outer loop optimization method, instead opting for an end-to-end training method.

## E   Additional Visualization

We have incorporated the visualization of the attention map of the BCAB module on the traffic dataset, as depicted in Figure 5. This visualization demonstrates that different time sequences have

distinct attention scores for different the same set of basis vectors. It can also be seen that the attention mechanisms exhibit significant similarities between the past and future sequences. This demonstrates the consistency in the base elements between the two.

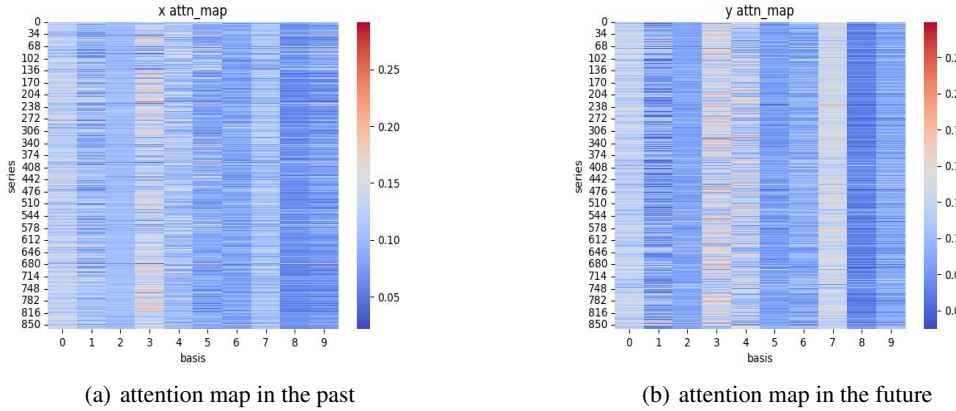

(a) attention map in the past                    (b) attention map in the future

Figure 5: corresponding set of attention maps from past and future perspectives

Additionally, we have provided a visualization of a specific time sequence alongside the features corresponding to the highest and lowest attention scores, as shown in Figure 6. The highest attention score is 0.2316, while the lowest attention score is 0.03371. Figure R2 highlights that the representation with a total of 8 sets of main peaks (Figure 6(c)) more comprehensively captures the patterns of the data compared to the configuration with only 2-3 main peaks (Figure 6(b)). This indicates a correlation between the attention scores and the relationship between time sequences and features.

It is important to note two key points. Firstly, since bases represent condensed patterns of time sequences, it is unlikely for a base to be identical to any single time sequence, especially when $N$ is small. Secondly, after the bases are processed through multiple linear and nonlinear layers in the network, they correspond to predicted sequences. Therefore, the numerical values of the bases serve as reference points only. The focus should be on the patterns exhibited by the bases.

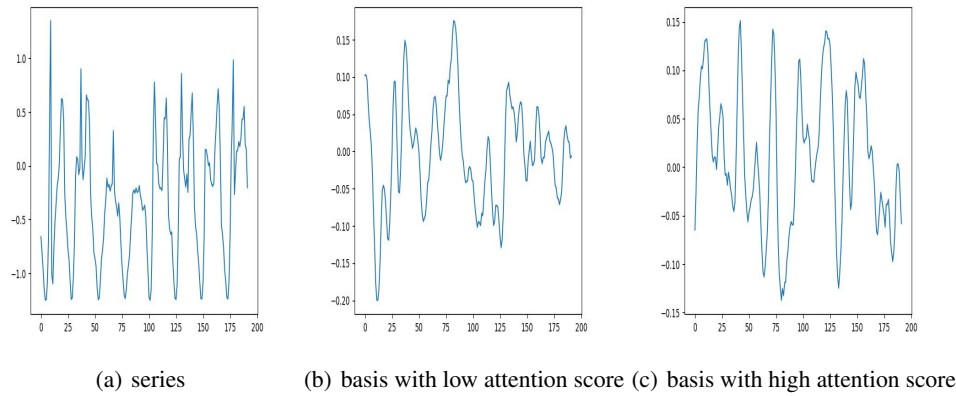

(a) series          (b) basis with low attention score (c) basis with high attention score

Figure 6: Time series and bases with higher and lower attention scores

## F  Analysis of the Model Complexity

Suppose that the input and output length in BasisFormer is $I$ and $O$ respectively when forecasting a single time series. Note that the time and space complexity of BasisFormer are of the same order. Therefore, we refer to both of them as complexity in the sequel.

With regards to the coef module, the complexity is primarily determined by the cross-attention mechanism. Within our approach, BCAB utilizes attention on the channel dimension, and we

encode the time sequence dimension to a specified hidden length $D_c \ll O$ via a linear layer during computation. Consequently, the complexity of this module is $\mathcal{O}(N)$, where $N$ is the number of bases - a fixed hyperparameter which is usually not large. In this step, we omit the number of BCAB stacks $M$, since $M$ is also a fixed hyperparameter. As previously mentioned in Appendix A.3, to limit overfitting, $M$ is typically set to 2.

The prediction module incorporates two Multilayer Perceptron (MLP) networks, which are employed for separating and concatenating different heads. Both MLP networks have bottlenecks with constant values, and they carry a complexity of $\mathcal{O}(O)$. In terms of the aggregation of different base vectors, the complexity also is $\mathcal{O}(O)$. Therefore, the cumulative complexity of this module is $\mathcal{O}(O)$.

In summary, the total complexity of our model is $\mathcal{O}(O)$. Table 18 provides a comparison of the computational complexity among different models, and BasisFormer achieves the lowest complexity among them.

Table 18: Comparison of computational complexity for different models.

| Methods | TIME | MEMORY |
|---|---|---|
| Fedformer | $\mathcal{O}(O)$ | $\mathcal{O}(O)$ |
| Autoformer | $\mathcal{O}(O \log O)$ | $\mathcal{O}(O \log O)$ |
| N-HiTS | $\mathcal{O}(O(1 - r^B)/(1 - r))$ | $\mathcal{O}(O(1 - r^B)/(1 - r))$ |
| FiLM | $\mathcal{O}(O)$ | $\mathcal{O}(O)$ |
| Dlinear | $\mathcal{O}(O)$ | $\mathcal{O}(O)$ |
| TCN | $\mathcal{O}(O)$ | $\mathcal{O}(O)$ |
| LogTrans | $\mathcal{O}(O \log O)$ | $\mathcal{O}(O^2)$ |
| Reformer | $\mathcal{O}(O \log O)$ | $\mathcal{O}(O \log O)$ |
| Informer | $\mathcal{O}(O \log O)$ | $\mathcal{O}(O \log O)$ |
| Basisformer | $\mathcal{O}(O)$ | $\mathcal{O}(O)$ |

