# OpenReview forum: "BasisFormer: Attention-based Time Series Forecasting with Learnable and Interpretable Basis"
_NeurIPS.cc/2023/Conference — NeurIPS 2023 poster_

### Official Review · Reviewer_yNQz · 2023-06-30

**Soundness:** 3 good
**Presentation:** 3 good
**Contribution:** 3 good
**Rating:** 8
**Confidence:** 5

**Summary:**

To be effective, a basis must be tailored to the specific set of time series data and exhibit distinct correlation with each time series within the set. As far as we know, the state-of-the-art methods are limited in their ability to satisfy both of these requirements simultaneously. To address this issue, the authors propose BasisFormer, an end-to-end time series forecasting architecture that leverages learnable and interpretable bases. Firstly, the authors acquire bases through adaptive self-supervised learning, which treats the historical and future sections of the time series as two distinct views and employs contrastive learning. Secondly, the authors design a Coef module that calculates the similarity coefficients between the time series and bases in the historical view via bidirectional cross-attention. Finally, the authors present a Forecast module that selects and consolidates the bases in the future view based on the similarity coefficients, resulting in accurate future predictions.

**Strengths:**

1. The authors acquire bases through adaptive self-supervised learning, which treats the historical and future sections of the time series as two distinct views and employs contrastive learning.
2. The authors design a Coef module that calculates the similarity coefficients between the time series and bases in the historical view via bidirectional cross-attention.
3. The authors present a Forecast module that selects and consolidates the bases in the future view based on the similarity coefficients, resulting in accurate future predictions.

**Weaknesses:**

1. The authors claim that this is a self-supervised learning model, and according to the reports of existing studies, the performance should be weaker than that of supervised models. I hope the authors can compare some time series prediction models based on self-supervised learning. Forgive me for being skeptical about the effect at the moment.

minor comment:

2. I would like the authors to be more clear about Eq. 7, why explainability is related to the smooth term, and to use the common l2-norm.


**Questions:**

The authors addressed all my concerns. So, I have no further comments.

**Limitations:**

Yes

---

> ### Author Rebuttal · Authors · 2023-08-10
>
> *Q1 - comparison with self-supervised methods*
>
> First, We would like to clarify that our model should not be classified as a self-supervised model in a strict sense. This is because our model utilizes a supervised loss function for predictions, while the alignment loss function and smoothing loss function are self-supervised. It is worth noting that our approach differs from classical self-supervised algorithms that involve a two-step process of self-supervised training followed by a supervised ridge regression. In contrast, we adopt an end-to-end training approach by integrating the self-supervised module with the prediction task.
>
> To address the reviewer's request, we have compared our model with several recent self-supervised methods, namely [1] and [2]. The comparative results are presented in the following table. For the experiments, we fix the input length at 96 and vary the output length from 96 to 720. **The table below illustrates that Basisformer outperforms Cost in terms of both MSE and MAE across all cases.** It is important to note that Cost is trained using a two-step process: self-supervised training followed by supervised ridge regression. In contrast, Basisformer trains the basis, coef, and forecast modules end-to-end. This enables Basisformer to learn a basis that is specifically tailored for forecasting, potentially leading to improved performance.
>
> | Models      |     |ours |        | Cost   |        |
> |:-----------:|:---:|:------:|:------:|:------:|:------:|
> | Metric      |   output_length  | MSE    | MAE    | MSE    | MAE    |
> | ETT         | 96  | 0.184  | 0.266  | 0.280  | 0.375  |
> |             | 192 | 0.248  | 0.307  | 0.480  | 0.506  |
> |             | 336 | 0.321  | 0.355  | 0.805  | 0.676  |
> |             | 720 | 0.410  | 0.404  | 1.562  | 0.955  |
> | electricity | 96  | 0.165  | 0.259  | 0.199  | 0.290  |
> |             | 192 | 0.178  | 0.272  | 0.199  | 0.292  |
> |             | 336 | 0.189  | 0.282  | 0.212  | 0.307  |
> |             | 720 | 0.223  | 0.311  | 0.246  | 0.338  |
> | exchange    | 96  | 0.085  | 0.205  | 0.263  | 0.393  |
> |             | 192 | 0.177  | 0.299  | 0.464  | 0.521  |
> |             | 336 | 0.336  | 0.421  | 0.833  | 0.691  |
> |             | 720 | 0.854  | 0.670  | 1.192  | 0.871  |
> | traffic     | 96  | 0.444  | 0.315  | 0.576  | 0.377  |
> |             | 192 | 0.460  | 0.316  | 0.546  | 0.359  |
> |             | 336 | 0.471  | 0.317  | 0.555  | 0.363  |
> |             | 720 | 0.486  | 0.318  | 0.591  | 0.379  |
> | weather     | 96  | 0.173  | 0.214  | 0.372  | 0.415  |
> |             | 192 | 0.223  | 0.257  | 0.528  | 0.517  |
> |             | 336 | 0.278  | 0.298  | 0.835  | 0.666  |
> |             | 720 | 0.355  | 0.347  | 1.394  | 0.894  |
> | illness     | 96  | 1.550  | 0.814  | 2.330  | 0.923  |
> |             | 192 | 1.516  | 0.819  | 2.497  | 0.962  |
> |             | 336 | 1.877  | 0.907  | 2.650  | 1.032  |
> |             | 720 | 1.878  | 0.902  | 2.829  | 1.062  |
>
>
>
>
> *Q2 - I would like the authors to be more clear about Eq. 7, why explainability is related to the smooth term, and to use the common l2-norm.*
>
> The incorporation of a loss term related to smoothness serves two significant purposes. **Firstly, it plays a crucial role in mitigating the risk of the learned basis fitting the noise present in the data, which would result in the emergence of high-frequency patterns.** If the basis is corrupted by noise, it becomes difficult to extract meaningful information such as trends or seasonality. Therefore, ensuring a smooth basis is essential for obtaining interpretable results.
>
> **Secondly, the inclusion of the smoothness loss term directly impacts the overall performance of the model.** This is evident from the results of ablation experiments involving different loss functions, as presented in Table 5. As our primary objective is long-range future forecasting, it is advantageous to have smoothly changing patterns in the basis.

---

> > ### Comment · Reviewer_yNQz · 2023-08-12
> > **Thanks to the author reply, which addressed all my concerns.**
> >
> > I have no further questions and I have improved my confidence score.

---

> > > ### Author Response · Authors · 2023-08-15
> > > **reply to reviewer yNQz**
> > >
> > > We extend our gratitude to the reviewer yNQz for offering meticulous comments and encouraging feedback. Your insightful suggestions have greatly contributed to enhancing the quality of our paper.

---

### Official Review · Reviewer_CQBS · 2023-07-06

**Soundness:** 3 good
**Presentation:** 3 good
**Contribution:** 2 fair
**Rating:** 4
**Confidence:** 5

**Summary:**

This paper addresses the problem of finding effective bases for time series forecasting models. Current methods are limited in their ability to satisfy the requirements of being tailored to specific time series data and exhibiting distinct correlation with each time series. To tackle this challenge, the authors propose BasisFormer, an end-to-end architecture that leverages learnable and interpretable bases. Bases are obtained through adaptive self-supervised learning, where historical and future time series sections are treated as distinct views and contrastive learning is used. The proposed architecture includes a Coef module that calculates similarity coefficients between time series and bases using bidirectional cross attention, and a Forecast module that selects and consolidates bases for accurate future predictions. Extensive experiments on six datasets demonstrate that BasisFormer outperforms previous methods for both univariate and multivariate forecasting tasks, achieving considerable improvements in performance.

**Strengths:**

This paper has studied a time series forecasting problem. The paper is well written. The significance of this work lies in addressing the limitations of existing methods in time series forecasting by proposing BasisFormer, an architecture that leverages learnable and interpretable bases. This approach allows for tailored modeling of specific time series data and distinct correlations with each time series, leading to improved forecasting accuracy.

**Weaknesses:**

However, I still have several concerns towards this paper. First, the illustration and motivation for the basis/bases is not very clear. What are the basis in essence? How can you define basis in time series? Are basis really needed for time series forecasting? What good effects can basis learning bring up? How to verify it? Second, in the main contribution, though the authors combine several techniques together, including self-supervised learning, contrastive learning, cross attention, basis selection. However, I don't find the necessity for each component. Third, the experimental results are less convincing. The proposed method adopts contrastive learning, but it does not compare the performance with other self-supervised learning methods, such as [1] [2]. Other works adopt periodic basis are also not discussed in related work [3].

[1] Cost: contrastive learning of disentangled seasonal-trend representations for time series forecasting. ICLR 2022.
[2] Time-Series Representation Learning via Temporal and Contextual Contrasting. IJCAI 2021.
[3] DEPTS: Deep Expansion Learning for Periodic Time Series Forecasting. ICLR 2022.

**Questions:**

see weakness

---

> ### Author Rebuttal · Authors · 2023-08-10
>
> *Q1 - motivation for baisis*
>
> As mentioned in Lines 20-22 on Page 1, bases are defined as **sequences that capture the underlying temporal patterns for a set of time series and serve as the key factors driving changes in the data over time**. They may encompass trends, seasonalities, and other vital elements that aid in modeling and forecasting time series data.
>
>
> Regarding the necessity of bases, **it is worth noting that nearly all time series forecasting models can be viewed as basis-driven models.** For example, Transformer-related models [8] [9] rely on covariate encoding, akin to bases, and some models explicitly depend on specific bases such as Fourier or Legendre bases [2] [3]. Experimental validation in this regard has been carried out previously, as exemplified by the following table citing experimental results from Dlinear [4]. In the presented experiments, the absence of bases (i.e., covariates) represented by timestamps (denoted as "wo/Temp") leads to a significant decrease in model performance, underscoring the importance of bases. It is important to note that the values in the table are presented in MSE format, where smaller values indicate better performance.
>
> | Methods    | Embedding | 96    | 192    | 336   | 720   |
> |:----------:|:---------:|:-----:|:------:|:-----:|:-----:|
> | Fedformer  | ALL       | 0.597 | 0.606  | 0.627 | 0.649 |
> |            | wo/Temp   | 0.613 | 0.623  | 0.65  | 0.677 |
> | Autoformer | ALL       | 0.629 | 0.647  | 0.676 | 0.638 |
> |            | wo/Temp   | 0.681 | 0.665  | 0.908 | 0.769 |
> | Informer   | ALL       | 0.719 | 0.696  | 0.777 | 0.864 |
> |            | wo/Temp   | 0.754 | 0.780  | 0.903 | 1.259 |
>
>
> The objective of Basisformer is to further show that learnable basis is preferred to manually specified basis for time series forecasting, and the usefulness of learnable basis is demonstrated in Table 5 (Page 8) and Table 6 (Page 9). In Table 5, by learning the basis adaptively from the data via the InfoNCE loss (cf. Section 3.3), our proposed Basisformer achieves an average performance improvement of 5.2%. Table 6 further supports this finding by showing that using a learnable basis instead of the manually specified covariates in Autoformer [8] and Fedformer [9] also leads to performance enhancement.
>
> *Q2 - contribution of contrastive learning and cross attention*
>
> As discussed in the second paragraph of Section I, the application of bases for time series forecasting involves three essential steps. Firstly, an appropriate basis is learned for the set of time series under consideration. Secondly, each time series in the set is decomposed based on the learned basis, which entails calculating coefficients or weights that represent the similarity or projection energy of the time series with respect to each vector in the basis. Finally, the prediction is determined by aggregating the future part of the basis using the computed weights.
>
> To this end, we employ self-supervised learning, specifically contrastive learning, to learn the basis, as described in Section 3.3. Cross attention is then utilized to perform basis projection or selection, which corresponds to the second step, as explained in Section 3.1.
>
> **To demonstrate the necessity of the self-supervised basis learning module, we conducted an ablation study (Lines 283-289, Page 7).** The results show that employing only the standalone self-supervised loss, namely the infoNCE loss, leads to an average performance improvement of 5.2%.
>
> Furthermore, we conducted additional experiments to validate the effectiveness of cross attention. In these experiments, we removed cross attention and basis selection by setting the number of stacked BCABs (Bidirectional Cross-Attention Blocks) in the model to 0. The experimental results are presented in the table below.
> | BCAB number | 0     | 1     | 2     | 3     | 4     |
> |-------------|-------|-------|-------|-------|-------|
> | Metric      | MSE   | MSE   | MSE   | MSE   | MSE   |
> | 96          | 0.186 | 0.166 | 0.166 | 0.168 | 0.171 |
> | 192         | 0.187 | 0.176 | 0.176 | 0.176 | 0.179 |
> | 336         | 0.208 | 0.187 | 0.190 | 0.190 | 0.191 |
> | 720         | 0.244 | 0.228 | 0.218 | 0.234 | 0.237 |
> | avg         | 0.206 | 0.189 | 0.187 | 0.192 | 0.194 |
>
> **The table highlights that, without altering other conditions, the absence of BCAB (i.e., cross attention for basis selection) resulted in an average performance decrease of 10.16% compared to the best-performing configuration with 2 layers of BCAB.** This emphasizes the effectiveness of cross attention in the model.
>
> *Q3 - comparison with self-supervised methods*
>
> Due to time constraints, we are currently able to present comparative results with only one method, Cost. For the experiments, we fix the input length at 96 and vary the output length from 96 to 720. **Table R1 in the PDF attached to the general response illustrates that Basisformer outperforms Cost in terms of both MSE and MAE across all cases.** It is important to note that Cost is trained using a two-step process: self-supervised training followed by supervised ridge regression. In contrast, Basisformer trains the basis, coef, and forecast modules end-to-end. This enables Basisformer to learn a basis that is specifically tailored for forecasting, potentially leading to improved performance.
>
> **Regarding DEPTS, we will discuss it in Section 2 as follows: **"DEPTS tackles the challenges posed by intricate dependencies and multiple periodicities in periodic time series through the implementation of a deep expansion learning framework. However, the complex initialization and optimization strategies employed by DEPTS, as well as its limitation of only being applicable to periodic sequences, have motivated us to develop a simpler and more universally applicable basis learning framework."

---

> ### Comment · Area_Chair_aCGC · 2023-08-19
> **Request for Reviewer CQBS to respond to authors' rebuttal**
>
> Reviewer CQBS, as there are only 2 days left in the author discussion period, would you please read the authors' response, explain the extent to which their answers address your concerns, and whether you will adjust your rating.
>
> If you decide to keep your score, please justify this decision, specifying which aspects of the paper or response have been the deciding factors in you keeping your score.

---

### Official Review · Reviewer_z6WM · 2023-07-11

**Soundness:** 2 fair
**Presentation:** 3 good
**Contribution:** 3 good
**Rating:** 4
**Confidence:** 4

**Summary:**

This paper studies basis learning for time series forecasting for which the past and future basis representations are aligned. Contrastive learning is used to build the time series basis and similarity between the past values and basis is used for time series prediction. The experiments on several time series forecasting datasets show improvement, especially for multivariate forecasting, when the input length (history) is short.


**Strengths:**

- The paper is overall well written
- The idea of using contrastive learning to learn basis is sound
- The proposed approach improves over baselines especially for multivariate case


**Weaknesses:**

- There is a strong emphasis on the consistency of the representations for past and future but this is directly experimented/evaluated
- Reproducibility is questionable without source code and scripts to run as the approach composed of several components and steps of training
- Proposed approach is particularly improves over baselines when the history to predict is shorter, however the motivation/need for a shorter history is not supported


**Questions:**

- Basisformer seems to have more advantages over baselines for the multivariate time series forecasting, how do you explain this?
- How models are trained for the longer input sequence? Is it only inference time change?
- Increasing the input length gives more advantage to the baselines. Do you conduct further analysis of the inference time versus performance for different length of input?
- Do you plan to share the implementation?


**Limitations:**

Yes

---

> ### Author Rebuttal · Authors · 2023-08-09
>
> *Q1 - past & future consistency*
>
> Thank you for bringing this to our attention. **We have shown the consistency of representations for the past and future sequences in our model in Figure R1 in the PDF file attached to the general response**, where the attention mechanisms exhibit significant similarities between the past and future sequences. This demonstrates the consistency in the base elements between the two.
>
> **Moreover, we have validated the significance of this past and future consistency in Table 5 (Page 8) and Table 6 (Page 9). **In Table 5, by ensuring such consistency through the InfoNCE loss, our proposed Basisformer achieves an average performance improvement of 5.2%. Table 6 further supports this finding by showing that consistency in Autoformer [8] and Fedformer [9] also leads to performance enhancement.
>
> *Q2 - reproducibility*
>
> As mentioned at the end of Abstract, full code of our model will be available at the time of publication.
>
> *Q3 - motivation for a short history*
>
> As mentioned in Lines 392-393 in the supplemtary material, "Throughout our research, we maintain consistency in our experimental settings by fixing the input length to be $96$ (with a reduced input length of $36$ for the illness dataset), instead of using a longer length. The main rationale behind this decision is that, in practical scenarios where the model is deployed as an online service and tasked with predicting a long range of the future at a granular level of minutes or hours, collecting a lengthy history (i.e., spanning 720 timestamps) for a large number of time series in real-time can be quite challenging. Therefore, **the adoption of an input length of 96 proves to be more practical and feasible."**
>
> Indeed this is a common practice in  N-beats[1], Autoformer[8], and Fedformer[9].
>
> On the other hand, to further demonstrate the superiority of Basisformer for longer input sequences, we have conducted experiments in Section A.2 in the supplementary material. As discussed in Lines 412-418, "concerning longer inputs, our method surpasses recent approaches such as Dlinear, FiLM, and N-HiTS, with an average MSE performance improvement of 1.35\%, 0.63\%, and 7.75\%, respectively, and a corresponding evaluation MAE performance improvement of 3.15\%, 2.33\%, and 4.06\%, respectively. It is noteworthy that our approach requires an input length of 192 (72 for the illness dataset), which is at least 40\% lower than the input length of the other three methods. **Furthermore, for even longer input lengths, our model's performance can be further enhanced, signifying that our approach can leverage limited data more efficiently." **
>
> *Q4 - performance is better for multivariate time series*
>
> The advantages of Basisformer over baselines for multivariate time series forecasting can be explained by considering the nature of multidimensional time series and the role of our model's basis.
>
> The basis functions in our model play a crucial role in capturing the primary patterns within the time series, akin to the principles of principle component analysis (PCA) to some extent. It is important to note that PCA performs better when the observed data are more correlated.
>
> Multidimensional time series, in comparison to single-dimensional ones, tend to have higher levels of correlation. For instance, traffic datasets often exhibit peaks during morning and evening hours, indicating correlated patterns. Consequently, the presence of such correlations allows our model to effectively learn the basis functions, leading to improved forecasting performance.
>
> **In summary, the utilization of basis functions in Basisformer, combined with the presence of correlated patterns in multidimensional time series, allows for better noise mitigation and identification of key patterns. **This ultimately enhances the overall performance of our approach for multivariate time series forecasting.
>
> *Q5 - how to train the model for longer inputs*
>
> Our model is trained using an end-to-end approach. When dealing with longer input sequences, we follow a similar methodology as N-hits [2] and Dlinear [4]. Specifically, we retrain the model using the extended input sequences along with their corresponding output sequences. Consequently, both the training and inference times are modified to accommodate the longer input sequences.
>
> *Q6 - time vs input length*
>
> We have provided a table below that displays the inference times for various input lengths, denoted as $I={96,192,336,720}$, and output lengths, denoted as $O={96,192,336,720}$. These measurements are based on the "exchange" dataset.
> |    | O=96     | O=192     | O=336     |     O=720      |
> |-------|-----------|-----------|-----------|-----------|
> | **I=96**  | 0.000833  | 0.001211  | 0.001419  | 0.002110  |
> | **I=192** | 0.000884  | 0.001285  | 0.001437  | 0.002139  |
> | **I=336** | 0.000893  | 0.001338  | 0.001469  | 0.002194  |
> | **I=720** | 0.000941  | 0.001364  | 0.001547  | 0.002246  |
>
> The table above illustrates the average inference time per instance of our algorithm under different configurations, measured in seconds. As observed, our algorithm exhibits notable speed, averaging at the millisecond level. Furthermore, when comparing the increase in output length to the extension of the input length, the additional inference time incurred by augmenting the input length is minimal. This is attributed to the preprocessing step where the input sequence, regardless of its length, is projected into a fixed-length (usually 100) sequence using a linear layer. **As a result, extending the input length does not significantly amplify the time consumption in our method.**

---

> ### Comment · Area_Chair_aCGC · 2023-08-19
> **Request for Reviewer z6WM to respond to authors' comments**
>
> Reviewer z6WM, as there are only 2 days left in the author discussion period, would you please read the authors' response, explain the extent to which their answers address your concerns, and whether you will adjust your rating.
>
> If you decide to keep your score, please justify this decision, specifying which aspects of the paper or response have been the deciding factors in you keeping your score.

---

### Official Review · Reviewer_twjH · 2023-07-25

**Soundness:** 3 good
**Presentation:** 3 good
**Contribution:** 3 good
**Rating:** 7
**Confidence:** 3

**Summary:**

This paper proposed BasisFormer which is an end-to-end time series forecasting model that leverages learnable and interpretable bases. BasisFormer treats the historical and future sections of the time series as two distinct views and using contrastive learning. By making use of Coef module and Forecast module, the BasisFormer outperforms previous state-of-the-art methods for univariate and multivariate forecasting tasks.

**Strengths:**

1. Contrastive learning objective is applied for basis learning which guarantees the consistency between the historical and future sections of the time series. And when applying the SSL module to other frameworks, there is a performance improvement of approximately 5%, which suggests the general application of the learnable basis.

2. Based on the experiment results on six datasets, the proposed BasisFormer model outperforms previous SOTA methods on univariate and multivariate forecasting tasks.

3. The network architecture of BasisFormer is carefully designed and well analyzed through ablation studies and model comparison. The paper is well written and easy to understand.


**Weaknesses:**

In 4.3, the author analyzed the interpretability of the learned bases by visualizing the time series and the corresponding learned basis. An additional visualization of the attention distribution in BCAB module can be helpful for checking different weights assigned to each basis in different attention heads, and thus understanding the network behavior.

**Questions:**

At this point, I don't have specific questions to ask. The paper is clearly written.

---

> ### Author Rebuttal · Authors · 2023-08-09
>
> *Q1 - Visualization*
>
> Thanks for pointing this out! We have incorporated the visualization of the attention map of the BCAB module on the traffic dataset, as depicted in Figure R1 in the PDF file attached to the global response. This visualization demonstrates that different time sequences have distinct attention scores for different the same set of basis vectors.
>
> Additionally, we have provided a visualization of a specific time sequence alongside the features corresponding to the highest and lowest attention scores, as shown in Figure R2. The highest attention score is 0.2316, while the lowest attention score is 0.03371. Figure R2 highlights that the representation with a total of 8 sets of main peaks (Figure R2(c)) more comprehensively captures the patterns of the data compared to the configuration with only 2-3 main peaks (Figure R2(b)). This indicates a correlation between the attention scores and the relationship between time sequences and features.
>
> It is important to note two key points. Firstly, since bases represent condensed patterns of time sequences, it is unlikely for a base to be identical to any single time sequence, especially when N is small. Secondly, after the bases are processed through multiple linear and nonlinear layers in the network, they correspond to predicted sequences. Therefore, the numerical values of the bases serve as reference points only. **The focus should be on the patterns exhibited by the bases.**

---

### Official Review · Reviewer_u9Fd · 2023-07-27

**Soundness:** 3 good
**Presentation:** 3 good
**Contribution:** 2 fair
**Rating:** 5
**Confidence:** 5

**Summary:**

The authors of the paper propose a time series forecasting architecture with a self-supervised method for basis learning, called BasisFormer. Their assumption is that the selection of basis for a time series is consistent across both historical and future sections of the time series. They introduce a Coef module that measures the similarity between the time series and the basis in the historical view via bidirectional cross-attention, and a Forecast module that consolidates vectors from the basis in the future view according to the similarity yielded by the Coef module. In their evaluation they demonstrate improvements in forecasting tasks.

**Strengths:**

1. The paper is well-written with clear explanations of the proposed architecture.

2. The empirical results, including the ones that were presented in the supplementary material, support the original claims across the manuscript.


**Weaknesses:**

1. In order to further validate the claims presented in this work, I would expect seeing another comparison to methods involve discretization of time-series, such as [Moskovitch, R. and Shahar, Y., 2015. Classification of multivariate time series via temporal abstraction and time intervals mining. Knowledge and Information Systems].

**Questions:**

1. What happens if bases are not learnable directly from the time-series, but require a domain expert's KB?


**Limitations:**

The novelty of this work is limited to cases where bases do exist and are detectable automatically in the input time-series, where in many cases the provided input is shorter than the length required for identifying cases of seasonality, for example.

---

> ### Author Rebuttal · Authors · 2023-08-09
>
> *Q1 – Comparison with time series discretization*
>
> We appreciate your suggestion to include a comparison with methods involving the discretization of time-series, such as Moskovitch and Shahar (2015). Unfortunately, we were unable to find the corresponding code and data for comparison. Therefore, we have chosen an alternative method for comparison, namely Boss [R1], which utilizes the Bag-Of-SFA-symbols method for feature extraction.
>
> In order to evaluate the effectiveness of our approach, we have conducted a classification task using several UCR datasets. The description of the datasets is as follows:
>
> | Dataset | Train Size | Test Size | Length | Classes | Type      | is_predictable | description                                       |
> |---------|------------|-----------|--------|---------|-----------|----------------|---------------------------------------------------|
> | Mallat  | 55         | 2345      | 1024   | 8       | SIMULATED | Y              | a simulated dataset                               |
> | Rock    | 20         | 50        | 2844   | 4       | SPECTRO   | Y              | rock examples from the ASTER spectral library     |
> | Phoneme | 214        | 1896      | 1024   | 39      | SOUND     | N              | Each series is extracted from the segmented audio |
> | FaceUCR | 200        | 2050      | 131    | 14      | IMAGE     | N              | rotationally aligned version of facial outline.   |
>
> To adapt our method for classification, we followed these steps:
>
> 1.	We partitioned the sequence into past and future parts, uniformly dividing them in a 6:4 ratio for all datasets. Different partitioning methods can be explored in future research to improve the model's performance.
>
> 2.	We used a self-supervised approach for training, reserving 10% of the original training data for validating self-supervised performance. The remaining data was used for training, and the self-supervised loss function included prediction, alignment, and smoothness losses. Early termination based on validation set performance was done with a patience of 3.
>
> 3.	From the well-trained self-supervised model, we extracted the aggregation coefficient matrix, specifically from the past perspective. This matrix was flattened to create sequence features, which were then fed into a random forest classifier for final classification. Notably, during self-supervised training, both past and future sequences were used for consistency, but only the past coefficient matrix was utilized in the classifier.
>
> We conducted a fair comparison by extracting features using both Boss and our model, ensuring that our feature parameter count did not exceed Boss's. We employed a random forest classifier with 100 features and a maximum depth of 30 for classification, and the results are summarized in the table below.
> |         | Boss        | Basisformer | Basisformer |
> |---------|-------------|-------------|-------------|
> |         | acc         | acc         | valid_loss  |
> | Mallat  | 0.83         | 0.87        | 0.12 |
> | Rock    | 0.56        | 0.72        | 0.18 |
> | Phoneme | 0.20    | 0.07        | 1.27|
> | FaceUCR | 0.68    | 0.41        | 1.68|
>
> **The applicability of our method to self-supervision relies on predictability and consistency between past and future data.** The validation set loss in the table indicates that datasets lacking predictability have high validation losses, posing challenges for loss function optimization.
> The Phoneme and FaceUCR datasets lack predictability. The Phoneme dataset includes speech segments from different individuals with random content before and after, while the FaceUCR dataset consists of flattened one-dimensional vectors of rotated face images, both lacking inherent predictability. These datasets require a holistic understanding of the entire sequence for meaningful interpretation, and as a result, our proposed Basisformer struggles to extract useful features, leading to lower performance compared to Boss.
>
> On the other hand, datasets like Mallat and Rock, exhibiting predictability and low validation losses, allow our approach to achieve superior performance over Boss. **Surprisingly, we achieve this performance using only the representation of the past sequence as input for the classifier.**
>
> *Q2- non-learnable basis*
>
> Our motivation in this research is to demonstrate the superiority of learnable bases compared to manually chosen bases. **We validated this by replacing the learnable basis with commonly used bases, such as sine-cosine encoding and covariate embedding**, in Table 3 (Page 8). However, we recognize that if a domain expert possesses comprehensive knowledge of the time series characteristics and designs a basis specifically for that set, the manually chosen basis may be comparable or even superior to a learnable basis.
>
> *Q3- input length shorter than a period*
>
> We respectfully disagree with the mentioned comment. In our research, the notion of the basis expands beyond the sole identification of seasonality. The basis serves as a condensed representation or summary of the inherent shape or pattern within a collection of time series data. As the basis is learnable, it can encompass any relevant information that aids in forecasting.
>
> It is important to note that our approach involves learning both the historical and future components of the basis from the training data. During inference, these learned components are fixed. **As a result, our forecasting process solely requires understanding how these components map to each other in order to predict the future.**
>
> [R1] Schäfer, Patrick., 2015. The BOSS is concerned with time series classification in the presence of noise. DMKD

---

> > ### Comment · Reviewer_u9Fd · 2023-08-15
> >
> > Thank you for the detailed response. I have improved my confidence score.

---

> > > ### Author Response · Authors · 2023-08-17
> > > **reply to reviewer u9Fd**
> > >
> > > We appreciate your suggestions once again, as they have provided valuable insights for improving our paper.

---

### Author Rebuttal · Authors · 2023-08-10

**General Response to All Reviewers**

We sincerely thank all the reviewers for their valuable suggestions. We are delighted by the unanimous recognition of our work and appreciate the reviewers' positive feedback on the carefully designed network architecture and the use of contrastive learning in BasisFormer.

We have thoroughly reviewed each of the reviewers' questions and suggestions, and we are grateful for their patience and diligence. In response, we have conducted additional experiments, introduced a new baseline, provided visualizations to address any lingering questions, and emphasized the significance of our work.

In the following rebuttal, we address each reviewer's comments individually. The reviewer’s comments are shown in italics. The paragraph(s) following them is the authors’ response. Unless otherwise specified, all references to pages, equations, sections, and citations refer to the original paper. Additionally, figures, tables, and citations prefixed with "R" (e.g., [R1]) are newly added citations in this rebuttal. All newly added images and a table are enclosed within a separate single-page PDF attached to this general response. We will incorporate the suggested revisions into the final camera-ready version to enhance the clarity and persuasiveness of our paper.

Once again, we would like to express our gratitude to the reviewers for their insightful feedback, which has helped us identify areas for improvement and refine our work. We welcome any further insights or concerns that would contribute to enhancing the paper according to the reviewers' perspectives.

---

### Decision · Program_Chairs · 2023-09-21

**Decision:**

Accept (poster)

**Comment:**

The paper presents a new method that leverages similarities between time series in a contrastive learning framework, with the main selling point being the considerable improvement in performance over state of the art methods. The reviewers praised the clarity of the writing, the explanations concerning the design decisions, the empirical results, and the analysis through ablation studies that yielded valuable insights. Three of the reviewers indicated acceptance, 2 of which gave very high scores. On the other hand, Reviewer z6WM had concerns about the consistency of the representations, the model performance on longer time series and reproducibility of the experiments. Meanwhile, Reviewer CQBS issued concerns about the motivation for bases and each of the components. The authors have, in my opinion, responded adequately to the concerns raised by Reviewers z6WM and CQBS. As these two reviewers have not read the authors' response, I will go with my own assessment and consider most of their comments addressed.
All things considered, I recommend acceptance of this paper.
I do, however, urge the authors to incorporate their responses on consistency and length of time series in the discussion part of their camera-ready paper, bolstering the explanations that they already have provided on these topics in the current manuscript.